# ROBUST GRAPH DIFFUSION MODEL

## ABSTRACT

Diffusion models represent a powerful class of generative models known for their solid theoretical foundations and remarkable performance across diverse tasks and domains. While diffusion models have been extensively utilized for generating entire graphs or small-scale graphs, no diffusion-based approaches have been developed to synthesize graph structures within an existing graph, including synthetic nodes and their associated edges. In this study, we introduce the Robust Graph Diffusion Model (RGDM), designed to generate labeled synthetic graph structures consisting of nodes and edges that integrate seamlessly into a given graph. The RGDM consists of a Robust Graph Autoencoder (RGAE) and a Latent Diffusion Model (LDM). Leveraging an edge selection mechanism and an innovative low-rank regularization on the latent feature, the RGDM produces clean and high-quality synthetic graph structures, even when trained on graphs subject to adversarial attacks. Comprehensive experimental evaluations reveal that Graph Neural Networks (GNNs) trained on the augmented graph, which is formed by merging the original attacked graph with the synthetic graph structures, exhibit significantly improved robustness against various graph adversarial attacks in the context of semi-supervised node classification. The code of the RGDM is available at https://anonymous.4open.science/r/RGDM.

## 1 INTRODUCTION

Diffusion models have achieved state-of-the-art performance in image generation (Ho et al., 2020; Gao et al., 2023a; Rombach et al., 2022; Baranchuk et al., 2022), and latent diffusion models (Rombach et al., 2022; Ho & Salimans, 2022) further extend this framework by incorporating conditioning signals into the denoising process to improve contextual fidelity. Motivated by these advances, recent studies (Trabucco et al., 2024; Azizi et al., 2023) propose using the synthetic data generated by diffusion models to augment the training set of deep neural networks. Some works (Niu et al., 2020; Song et al., 2021; Haefeli et al., 2022; Vignac et al., 2023; Limnios et al., 2023) also extend diffusion models to the graph-structured data, while they primarily focus on graph-level augmentation. Most existing node-level graph augmentation methods focus on increasing the number of labeled nodes by simple augmentation methods such as node-level mix-up (Han et al., 2022; Wang et al., 2021; Verma et al., 2021; Zhao et al., 2024; Jeong et al., 2024; Lu et al., 2024) and generating synthetic labeled nodes and edges by Generative Adversarial Networks (GANs) for imbalanced graph data (Jia et al., 2023a; Wu et al., 2023; Wang et al., 2018; Liang et al., 2020; Yang et al., 2019). We note that the node-level graph augmentation method, DoG (Wang et al., 2025), is not designed for robust graph learning. Although widely used in augmenting graph data, the computer vision literature (Dhariwal & Nichol, 2021) has demonstrated that GANs often exhibit instability during training and that the distribution of synthetic data generated by GANs poorly matches the distribution of real data.

As a result, the challenge of generating synthetic graph structures (SGS) within a given graph, which are the synthetic nodes and their corresponding edges, in a robust manner to enhance node classification under attacks, remains underexplored. In this work, we tackle the problem of node-level graph generation and propose the Robust Graph Diffusion Model (RGDM), designed to synthesize high-quality graph structures within a single graph, even under adversarial attacks. These synthetic structures include both synthetic nodes and edges connecting them within the same graph, forming an augmented graph as illustrated in Figure 1. Robust GNNs will be trained on the augmented graph, and superior performance on the augmented graph is expected due to the enlarged set of labeled nodes in the augmented graph. Generating high-quality SGS in adversarial scenarios for improved performance is challenging, important, and unaddressed in the robust graph learning literature. To address this challenge, RGDM integrates a Robust Graph Autoencoder (RGAE) with

a Latent Diffusion Model (LDM). The RGAE incorporates an edge selection mechanism and a low-rank regularization on the latent features to counteract adversarial perturbations in the attacked graph. The low-rank regularization is motivated by the Low Frequency Property (LFP) (Rahaman et al., 2019; Arora et al., 2019; Cao et al., 2021; Choraria et al., 2022; Wang et al., 2024; 2025), which indicates that the low-rank part of the latent features covers the dominant information in the ground truth label while learning only a small portion of the adversarial noise, as illustrated in Figure 4 in Section F.1. The low-rank regularization is further justified by our novel theoretical result in Theorem 3.1, which shows that the low-rank feature ensures a smaller kernel complexity (KC) and thus contributes to a smaller generalization bound for transductive node classification.

Figure 1 illustrates the 2-hop neighborhood of a real node and a synthetic node that have similar node features, evidenced by the t-SNE visualization in Figure 5 in Section F.9 of the appendix. The fidelity of the SGS generated by the RGDM is proved in Table 8 and Table 9 in Section F.3 of the appendix, where the synthetic nodes and edges in the SGS generated by the RGDM exhibit similar Frechet Node Distance (FND) score and Frechet Edge Distance (FED) scores as the nodes and edges in the original clean graph. The FND score and the FED score, defined in Section F.3 of the appendix, measure the quality of the synthetic nodes and edges compared to the real nodes and edges in the original clean graph, with lower FND and FED scores indicating better quality. As shown in Table 8 and Table 9, the SGS generated by RGDM exhibits significantly lower FND and FED scores compared to the SGS generated by the baseline diffusion model DDPM (Ho et al., 2020) and various GAN-based SGS generation methods (Qu et al., 2021; Gao et al., 2023b; Zhao et al., 2021a), demonstrating the advantages of the RGDM in generating faithful synthetic graph structures. We remark that since the synthetic graph structures generated by the RGDM are faithful

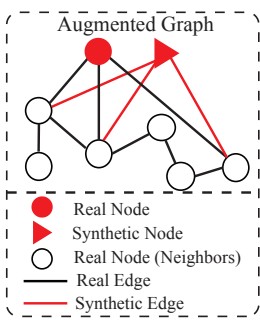

Figure 1: Illustration of the 2-hop neighborhood of a real node and a synthetic node in the augmented graph. Figure 5 in Section F.9 illustrates the t-SNE visualization of the augmented graph.

and similar to nodes and edges in the original clean graph, incorporating the synthetic graph structures into the original attacked graph dilutes the adversarial noises in it. For example, the RGDM generates 785 synthetic edges for the Citeseer dataset under the Metattack (Zügner & Günnemann, 2019) with a perturbation rate of 25%. After incorporating the faithful synthetic graph structures generated by the RGDM into the original attacked graph, the perturbation rate is reduced to roughly 20.6% in the augmented graph. The dilution of the adversarial noise provides an explanation of how the RGDM significantly improves the robustness of Graph Neural Networks (GNNs) trained on the augmented graph, as shown in Section 4.2.

**Difference from Existing Diffusion-based Purification Methods.** Recent works have explored applying diffusion models for data purification (He et al., 2025; Luo et al., 2025; Zhao et al., 2025; Xie et al., 2025; Chen et al., 2023), including graph purification (He et al., 2025; Luo et al., 2025; Zhao et al., 2025). However, the existing diffusion-based purification methods can not synthesize low-noise and faithful SGS. This is because the current diffusion-based graph purification methods (He et al., 2025; Luo et al., 2025; Zhao et al., 2025) are only for graph-level purification, and they are trained on the attacked graphs and the corresponding clean graphs to learn the denoising process. Because there are no clean graph data corresponding to the SGS, all the current graph-level, diffusion-based graph purification methods, such as (He et al., 2025; Luo et al., 2025), cannot be trained to purify the augmented graphs, where the augmented graph includes the original attacked graph and the SGS. Furthermore, although one can generate synthetic nodes/edges by vanilla diffusion-based methods, these generated SGS inevitably suffer from noise since the training data, that is, the original attacked graphs, contains adversarial noise. In contrast, our proposed RGDM does not need clean graphs in its training and inference process.

**Contributions.** The contributions of this paper are presented as follows.

First, we introduce a novel Robust Graph Diffusion Model (RGDM), the first node-level graph diffusion model designed to synthesize labeled graph structures for robust graph learning. Decoding a node's neighbors over the entire graph incurs quadratic complexity, as each node may connect to any other node. To address this issue, our RGDM also introduces a novel Sparse Hierarchical Edge Decoder (SHED), which reconstructs the edges connected to a node in an efficient sparse and hierarchical manner. Training GNNs on the augmented graph that integrates the synthetic graph structures

into the original graph leads to substantial improvements in semi-supervised node classification under graph adversarial attacks. As evidenced in Section F.3 and Section F.10 of the appendix, RGDM generates more faithful SGS than existing graph augmentation methods, and the GNNs trained on SGS demonstrate stronger robustness against adversarial attacks.

Second, benefiting from the new low-rank regularization on the latent feature in the RGAE, RGDM generates high-quality synthetic graph structures that closely resemble the original clean graph, even when trained on graphs compromised by adversarial attacks. Table 8 and Table 9 in the appendix evidence the quality of the synthetic graph structures, as measured by the FND and FED scores defined in Section F.3 of the appendix. The low-rank regularization in the RGAE is empirically inspired by LFP and theoretically inspired by our novel theoretical result in Theorem 3.1, which demonstrates that a low-rank feature leads to reduced kernel complexity (KC), thereby contributing to a tighter generalization bound for transductive node classification. The augmented graph, formed by integrating the synthetic graph structures into the attacked graph, effectively reduces the impact of adversarial noise compared to the original attacked graph. Consequently, GNNs trained on this augmented graph exhibit enhanced robustness. As shown in Section 4.2, existing robust GNNs, such as Pro-GNN (Jin et al., 2020) and SG-GSR (In et al., 2024), trained on the augmented graph by RGDM achieve state-of-the-art performance under various types of graph adversarial attacks.

## 2 RELATED WORKS

**Data Augmentation for Graph Learning.** To improve the performance of GNNs, node-level graph data augmentation methods have been studied to modify the structure (Gasteiger et al., 2019; Zhao et al., 2021b; Rong et al., 2020; Feng et al., 2022; Lai et al., 2024), features (You et al., 2020; Kong et al., 2022; Azad & Fang, 2024), or node labels (You et al., 2020; Kong et al., 2022; Azad & Fang, 2024). Studies have demonstrated that increasing the number of labeled nodes based on Mix-up strategies (Han et al., 2022; Wang et al., 2021; Verma et al., 2021; Zhao et al., 2024; Jeong et al., 2024; Lu et al., 2024) can greatly enhance the performance of the GNNs. Recent works address node and edge imbalance by generating synthetic nodes and edges (Qu et al., 2021; Zhao et al., 2021a; Hsu et al., 2024; Gao et al., 2023b; Hsu et al., 2023), but predominantly rely on Generative Adversarial Networks (GANs). However, the computer vision literature (Dhariwal & Nichol, 2021) has demonstrated that GANs often exhibit instability during training and that the distribution of synthetic data generated by GANs poorly matches the distribution of real data. Although diffusion models have demonstrated superior capability in generating faithful graph data, all existing graph diffusion models (Song et al., 2021; Niu et al., 2020; Haefeli et al., 2022; Vignac et al., 2023; Limnios et al., 2023) are designed for graph-level generation and lack the capability to generate synthetic nodes along with their associated edges within an existing graph. In contrast, our work aims to simultaneously generate faithful synthetic nodes and their associated edges within a given graph using the diffusion model.

**Graph Adversarial Attacks and Defense.** Despite the success of GNNs in graph-structured tasks (Kipf & Welling, 2017; Zhang & Chen, 2018), recent studies reveal their vulnerability to adversarial attacks (Dai et al., 2022). Depending on the attack objective, threat models are categorized as: targeted attacks (Zügner et al., 2018), which mislead predictions on specific nodes, and untargeted attacks (Zügner & Günnemann, 2019; Sun et al., 2020), which reduce overall model accuracy. To defend against these threats, robust learning strategies are grouped into three categories: adversarial training, graph processing, and model robustification. Adversarial training (Li et al., 2022a; Feng et al., 2019) augments training data with crafted adversarial samples. Graph processing (Wu et al., 2019; Entezari et al., 2020; Jin et al., 2020; Lei et al., 2022) seeks to denoise perturbed graphs. Model robustification (Xie et al., 2023; Rusch et al., 2022; Song et al., 2022; Zhao et al., 2023; Jia et al., 2023b; Liu et al., 2024; Abbahaddou et al., 2024; In et al., 2024) enhances model resilience.

## 3 METHODS

### 3.1 PRELIMINARIES OF THE ATTRIBUTED GRAPH

An attributed graph consisting of $N$ nodes is represented by $\mathcal{G} = (\mathcal{V}, \mathbf{X}, \mathbf{A})$, where $\mathcal{V} = \{v_1, v_2, \ldots, v_N\}$ and $\mathcal{E} \subseteq \mathcal{V} \times \mathcal{V}$ denote the nodes and edges respectively. $\mathbf{X} \in \mathbb{R}^{N \times D}$ are the node attributes. $\mathbf{X}_i \in \mathbb{R}^D$, the $i$-th row of $\mathbf{X}$, denotes the attributes of the $i$-th node. $\mathbf{A} \in \{0, 1\}^{N \times N}$ is the adjacency matrix of the graph $\mathcal{G}$. $\tilde{\mathbf{A}} = \mathbf{A} + \mathbf{I}$ is the adjacency matrix of a graph with self-loops added. $\tilde{\mathbf{D}}$ denotes the diagonal degree matrix of $\tilde{\mathbf{A}}$. We use $\mathcal{N}(i) = \{j \mid \tilde{\mathbf{A}}_{i,j} = 1\}$ to denote the set

of the indexes of nodes connected to the node $v_i$. $\mathbf{Z} \in \mathbb{R}^{N \times D'}$ are the latent features of all the nodes in the graph, where $D'$ is the latent dimension. Let $\mathcal{V}_{\mathcal{L}}$ and $\mathcal{V}_{\mathcal{U}}$ denote the set of labeled nodes and unlabeled test nodes, respectively, and $|\mathcal{V}_{\mathcal{L}}| = m$, $|\mathcal{V}_{\mathcal{U}}| = u$. Let $Y_L = \{y_i | v_i \in V_L, y_i \in [C]\}$ denote the labels of nodes in $\mathcal{V}_{\mathcal{L}}$, where $C$ is the number of classes. Let $\mathbf{Y} = [\mathbf{y}_1; \mathbf{y}_2; \dots \mathbf{y}_N] \in \mathbb{R}^{N \times C}$ be the ground truth label matrix of all the nodes in $\mathcal{G}$, where $\mathbf{y}_i$ is the one-hot label vector of node $v_i$. Let $\mathbf{u} \in \mathbb{R}^N$ be a vector, we use $[\mathbf{u}]_{\mathcal{A}}$ to denote a vector formed by elements of $\mathbf{u}$ with indices in $\mathcal{A}$ for $\mathcal{A} \subseteq [N]$. If $\mathbf{u}$ is a matrix, then $[\mathbf{u}]_{\mathcal{A}}$ denotes a submatrix formed by rows of $\mathbf{u}$ with row indices in $\mathcal{A}$. $\|\cdot\|_{\mathrm{F}}$ denotes the Frobenius norm of a matrix, and $\|\cdot\|_p$ denotes the $p$-norm of a vector.

## 3.2 ROBUST GRAPH AUTOENCODER (RGAE)

In this section, we propose the Robust Graph Autoencoder (RGAE) capable of encoding node attributes and associated edges into continuous latent features and decoding these features to reconstruct the node attributes and edges in the graph. Beyond reconstruction, the RGAE is designed to be robust to adversarial perturbations in the node attributes and the edges in the graph, enhancing the resilience against potential graph adversarial attacks.

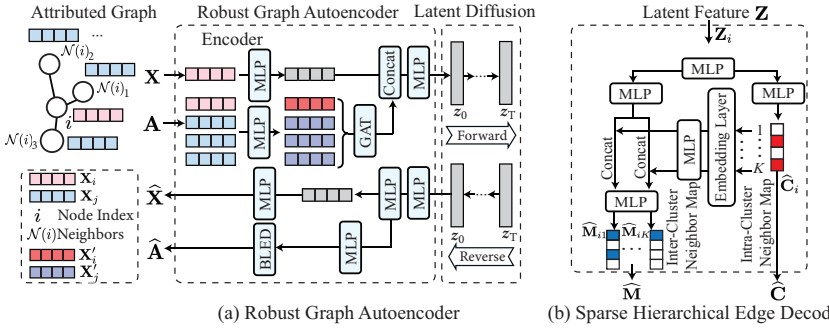

(a) Robust Graph Autoencoder      (b) Sparse Hierarchical Edge Decoder

Figure 2: Figure (a) illustrates the synthetic node generation process in Robust Graph Diffusion Model. Figure (b) illustrates the structure of the Sparse Hierarchical Edge Decoder (SHED).

**Robust Graph Encoder with Edge Selection.** We generate latent features by combining embeddings of each node's attributes $\mathbf{X}_i$ and those of its neighbors $\mathbf{X}_j \mid j \in \mathcal{N}(i)$. To incorporate edge information, we add positional embeddings to each neighbor's features as $\mathbf{X}'_j = \mathbf{X}_j + \mathrm{pos}(j)$ for $j \in \mathcal{N}(i)$, which are then aggregated using GAT layers (Veličković et al., 2018). GAT compute attention weights $\mathbf{B} \in \mathbb{R}^{N \times N}$ as $\mathbf{B}_{ij} = \mathrm{softmax}\left(\mathbf{A}_{ij}\sigma\left(\mathbf{X}'_i\mathbf{Q}\|\mathbf{X}'_j\mathbf{Q}\right)\mathbf{V}\right)$, where $\|$ denotes concatenation, $\sigma$ is the sigmoid function, $\mathbf{Q} \in \mathbb{R}^{D \times M}$, and $\mathbf{V} \in \mathbb{R}^{2M \times 1}$ are learnable weights.

However, if the graph is adversarially attacked, the graph may contain edges that propagate harmful information and distort the node representations. To address such challenge, inspired by graph sparsification for robustness (Jin et al., 2020; Zheng et al., 2020), we propose an edge selection method to purify the edges in the graph during the training of the RGAE. A binary decision mask $\mathbf{S} \in \{0, 1\}^{N \times N}$ retains only faithful edges. To enable gradient-based optimization, we relax $\mathbf{S}$ into a continuous approximation $\widehat{\mathbf{S}} \in (0, 1)^{N \times N}$ using the binary Gumbel-Softmax (Verelst & Tuytelaars, 2020; Bengio et al., 2013), where $\widehat{\mathbf{S}}_{ij} = \sigma\left(\frac{\theta_{ij}}{\tau}\right)$, where $\boldsymbol{\theta} \in \mathbb{R}^{N \times N}$ and $\tau$ is the temperature. $\boldsymbol{\theta}_{ij}$ is obtained by applying a linear layer to $\mathbf{X}'_i\|\mathbf{X}'_j$. The GAT output with edge selection is computed as $\mathbf{Z}'_i = \mathrm{ReLU}\left(\sum_{j \in \mathcal{N}(i)} \mathbf{S}_{ij}\mathbf{B}_{ij}\mathbf{X}'_i\mathbf{W}\right)$, where $\mathbf{W}$ is the learnable transformation matrix. We stack two GAT layers to obtain $\mathbf{Z}'_i$, capturing neighborhood and edge information of node $v_i$. The final latent feature for node $v_i$ is $\mathbf{Z}_i = \mathbf{Z}'_i\|f(\mathbf{X}_i)$, where $f(\cdot)$ is a multilayer perception (MLP).

**Sparse Hierarchical Edge Decoder (SHED).** Decoding edges from latent features can be computationally intensive, as each node may connect to any other node. To mitigate this, we propose a Sparse Hierarchical Edge Decoder (SHED), which decodes each latent feature into an inter-cluster and an intra-cluster neighbor map, as illustrated in Figure 2 (b). The SHED begins by partitioning the nodes into $K$ balanced clusters using balanced $K$-means clustering (Malinen & Fränti, 2014) based on node attributes. Each cluster has a maximum capacity of $M = \lceil \frac{N}{K} \rceil$. Within each cluster, nodes are indexed by their original graph order. For a node $v_i$, SHED first reconstructs an inter-

cluster neighbor map $\mathbf{C} \in \{0,1\}^{N \times K}$, where $\mathbf{C}_{ik} = 1$ if $v_i$ is connected to at least one node in cluster $k$. It then reconstructs the intra-cluster neighbor map as a tensor $\mathbf{M} \in \{0,1\}^{N \times K \times M}$, where $\mathbf{M}_{ikm} = 1$ indicates that $v_i$ is connected to the $m$-th node in cluster $k$. The SHED begins by computing the inter-cluster neighbor assignment $\widehat{\mathbf{C}}_i$ for each node $v_i$ using a single-layer MLP. The active cluster indices, $\mathcal{C}(i) = \{k \in [K] \mid \mathbf{C}_{ik} = 1\}$, are extracted and passed through a class-conditional embedding module $g(\cdot)$, which consists of a text encoder followed by a MLP, inspired by the classifier-free guidance technique (Ho & Salimans, 2022). This produces a set of embeddings $\mathcal{Z}(i) = \{g(k) \in \mathbb{R}^{D'} \mid k \in \mathcal{C}(i)\}$. Each embedding $g(k)$ is then concatenated with the latent representation $\mathbf{Z}_i$ from the alternate branch, and the result is processed by a decoding MLP, $g'(\cdot)$, to estimate the intra-cluster neighbor relation as $\widehat{\mathbf{M}}_{ik} = g'(\mathbf{Z}_i \| g(k))$. Compared to the Bi-Level Neighborhood Decoder (BLND) in DoG (Wang et al., 2025), the reconstructed neighbor map $\widehat{\mathbf{M}}$ is sparse for robust graph learning, which is enforced by the training of the RGAE detailed later.

**Optimization of the RGAE.** RGAE incorporates an edge selection mechanism in the encoder that gradually removes the adversarial edges. By learning an edge selection mask $\mathbf{S}$, the adjacency matrix of the graph is purified as $\mathbf{A} \circ \mathbf{S}$. By integrating the edge selection mechanism with SHED, we decompose $\mathbf{S}$ into two edge selection masks, $\mathbf{S}_{\text{inter}} \in \{0,1\}^{N \times K}$ and $\mathbf{S}_{\text{intra}} \in \{0,1\}^{N \times K \times M}$, for edge selection on the inter-cluster neighbor map $\mathbf{C}$ and intra-cluster neighbor map $\mathbf{M}$. At the end of each epoch, $\mathbf{S}_{\text{inter}}$ and $\mathbf{S}_{\text{intra}}$ are computed from $\mathbf{S}$. In particular, $[\mathbf{S}_{\text{inter}}]_{ik} = 1$ if and only if there is a node $v_j$ from the cluster $k$, such that $\mathbf{S}_{ij} = 1$. $[\mathbf{S}_{\text{intra}}]_{ikm} = 1$ if and only if $\mathbf{S}_{ij} = 1$, where $v_j$ is the $m$-th node in the cluster $k$.

In addition, we propose to mitigate the negative effects of the adversarial noise by the learned feature kernel to be low-rank. The kernel gram matrix $\mathbf{K}$ of the latent representations $\mathbf{Z} \in \mathbb{R}^{N \times D'}$ is calculated by $\mathbf{K} = \mathbf{Z}^\top \mathbf{Z} \in \mathbb{R}^{N \times N}$, where $D'$ is the dimension of the latent representation. Let $\left\{\widehat{\lambda}_i\right\}_{i=1}^n$ with $\widehat{\lambda}_1 \geq \widehat{\lambda}_2 \ldots \geq \widehat{\lambda}_{\min\{N,d\}} \geq \widehat{\lambda}_{\min\{N,d\}+1} = \ldots, = 0$ be the eigenvalues of $\mathbf{K}$. In order to encourage the features $\mathbf{Z}$ or the gram matrix $\mathbf{K} = \mathbf{Z}^\top \mathbf{Z}$ to be low-rank, we explicitly add the truncated nuclear norm (TNN), which is $\|\mathbf{K}\|_{r_0} := \sum_{r=r_0+1}^N \widehat{\lambda}_i$, to the training loss the RGAE. The starting rank $r_0 < \min(N, D')$ is the rank of the kernel gram matrix of the features we aim to obtain with the encoder of the RGAE, that is, if $\|\mathbf{K}\|_{r_0} = 0$, then $\text{rank}(\mathbf{K}) = r_0$. Therefore, the overall loss function of the RGAE is

$$\mathcal{L}_{\text{RGAE}}^{(t)}(\mathbf{S}_{\text{intra}}^{(t-1)}, \mathbf{S}_{\text{inter}}^{(t-1)}) = \mathcal{L}_{\text{C}}^{(t)} + \tau_{\text{S}}\mathcal{L}_{\text{S}}^{(t)} + \tau_{\text{T}}\|\mathbf{K}\|_{r_0}, \tag{1}$$

$$\mathcal{L}_{\text{C}}^{(t)} = \underbrace{\left\|\mathbf{X} - \widehat{\mathbf{X}}\right\|_2^2}_{\text{Node Reconstruction}} + \underbrace{\left(\left\|\mathbf{C} - \widehat{\mathbf{C}}\right\|_2^2 + \left\|\mathbf{M} - \widehat{\mathbf{M}}\right\|_2^2\right)}_{\text{Hierarchical Edge Reconstruction}}, \mathcal{L}_{\text{S}}^{(t)} = \left\|\mathbf{C} \circ \mathbf{S}_{\text{inter}}^{(t-1)} - \widehat{\mathbf{C}}\right\|_2^2 + \left\|\mathbf{M} \circ \mathbf{S}_{\text{intra}}^{(t-1)} - \widehat{\mathbf{M}}\right\|_1.$$

Here $\circ$ denotes the Hadamard product. $\mathcal{L}_{\text{RGAE}}^{(t)}$ is the overall training loss of the RGAE at the $t$-th epoch, $t \in [t_{\text{train}}]$ is the current epoch number and $t_{\text{train}}$ is the total number of training epochs. We initialize $\mathbf{S}_{\text{intra}}^{(0)} = \mathbf{C}$ and $\mathbf{S}_{\text{inter}}^{(t-1)} = \mathbf{M}$ before the first epoch of the training. $\tau_{\text{S}}$ and $\tau_{\text{T}}$ are the scale factors for the sparsification loss and the TNN. $\mathcal{L}_{\text{C}}^{(t)}$ and $\mathcal{L}_{\text{S}}^{(t)}$ are the consistency loss and the sparsification loss at the $t$-th epoch. The starting rank $r_0$ is selected by cross-validation as described in Section 4.1. The training of RGDM is described in Algorithm 1 in Section C of the appendix.

The node reconstruction loss and the hierarchical edge reconstruction loss ensure consistency between the reconstructed features and adjacency matrix and their inputs. The sparsification loss is motivated by prior work on graph structure learning for robust GNNs (Jin et al., 2020; Zheng et al., 2020), which aims to purify noisy or perturbed graph structures. he term $\left\|\mathbf{M} \circ \mathbf{S}_{\text{intra}}^{(t-1)} - \widehat{\mathbf{M}}\right\|_1$ in $\mathcal{L}_{\text{S}}^{(t)}$ ensures the sparsity of $\widehat{\mathbf{M}}$. Although the BLND in DoG (Wang et al., 2025) also performs edge reconstruction in a two-level manner, it is trained to precisely recover all edges connected to each node in the given graph, including those corrupted by adversarial attacks. In contrast, our SHED reconstructs a sparse and robust graph structure by selectively preserving the clean edges, motivated by prior studies showing that graph sparsification enhances the robustness of GNNs against adversarial attacks by removing noisy connections (Jin et al., 2020; Lei et al., 2022; Zheng et al., 2020; Chowdhury et al., 2024). Moreover, existing studies on adversarial defense reveal that perturbations typically reside in high-rank residual components, whereas clean signals lie in low-rank

subspaces (Awasthi et al., 2020; 2021). Projecting representations onto low-rank subspaces (Awasthi et al., 2020; 2021; Xu et al., 2024) or enforcing low-rank weight matrices (Savostianova et al., 2023) has been proven to be effective for enhancing robustness. However, research in graph adversarial defense has focused predominantly on reducing the rank of the adjacency matrix (Wu et al., 2022; Entezari et al., 2020; Deng et al., 2022; Zhou et al., 2025; Zhang et al., 2025). In contrast, our proposed low-rank regularization enforces the latent representation to be low-rank with a principled theoretical guarantee in Theorem 3.1.

**Motivation of Low-Rank Regularization.** We study how the information of the ground-truth class label and the adversarial noise are distributed on different eigenvectors of the feature gram matrix $\mathbf{K} = \mathbf{Z}^\top \mathbf{Z}$ by performing eigen-projection, where $\mathbf{Z}$ is the latent feature learned on the graph with the adversarial noise. We first compute the eigenvectors $\mathbf{U}$ of the feature gram matrix $\mathbf{K}$. Let $\mathbf{U}^{(1:r)} \in \mathbb{R}^{N \times r}$ be the top $r$-eigenvectors of $\mathbf{K}$ and $\mathbf{U}^{(r)}$ be the $r$-th eigenvector of $\mathbf{K}$. Then, the eigen-projection value of the ground-truth label $\mathbf{Y}$ on $\mathbf{U}^{(r)}$ is computed by $p_r = \frac{1}{C} \sum_{c=1}^C \left\| \mathbf{U}^{(r)\top} \mathbf{Y}^{(c)} \right\|_2^2 / \left\| \mathbf{Y}^{(c)} \right\|_2^2$ for $r \in [N]$, where $\mathbf{Y}^{(c)}$ is the $c$-th column of $\mathbf{Y}$. We let $\mathbf{p} = [p_1, \ldots, p_N] \in \mathbb{R}^N$. The eigen-projection $p_r$ reflects the amount of the signal projected onto the $r$-th eigenvector of $\mathbf{K}$, and the signal concentration ratio of a rank $r$ reflects the proportion of signal projected onto the top $r$ eigenvectors of $\mathbf{K}$. The signal concentration ratio for rank $r$ is computed by $\|\mathbf{p}(1:r)\|_1$, where $\mathbf{p}(1:r)$ contains the first $r$ elements of $\mathbf{p}$. It is observed from the red curves in the left of Figure 3 and Figure 4 in Section F.1 of the appendix that the projection of the ground truth labels mostly concentrates on the top eigenvectors of $\mathbf{K}$, known as the Low Frequency Property (LFP) widely studied in deep learning (Rahaman et al., 2019; Arora et al., 2019; Cao et al., 2021; Choraria et al., 2022; Wang et al., 2024; 2025). On the other hand, we study how the information of the adversarial noise is distributed on the eigenvectors of the feature gram matrix $\mathbf{K}$.

Let $\mathbf{Z}^{(\text{clean})}$ be the latent features obtained on the clean input graph without adversarial noise, then $\Delta\mathbf{Z} = \mathbf{Z} - \mathbf{Z}^{(\text{clean})}$ is the adversarial noise in the feature due to the attacks. The low-rank part of $\mathbf{Z}$ characterized by its top-$r$ eigenvectors is $\mathbf{Z}^{(\text{low-rank})} = \mathbf{U}^{(1:r)}\mathbf{U}^{(1:r)\top}\mathbf{Z} = \mathbf{U}^{(1:r)}\mathbf{U}^{(1:r)\top}(\mathbf{Z}^{(\text{clean})} + \Delta\mathbf{Z})$. Then $\mathbf{U}^{(1:r)}\mathbf{U}^{(1:r)\top}\mathbf{Z}^{(\text{clean})}$ is the low-rank part of the clean latent feature $\mathbf{Z}^{(\text{clean})}$, and $\mathbf{U}^{(1:r)}\mathbf{U}^{(1:r)\top}\Delta\mathbf{Z}$ is the adversarial noise induced by the adversarial attack in $\mathbf{Z}^{(\text{low-rank})}$. We compute eigen-projection value of the adversarial noise by $p_r^{(\Delta\mathbf{Z})} =$

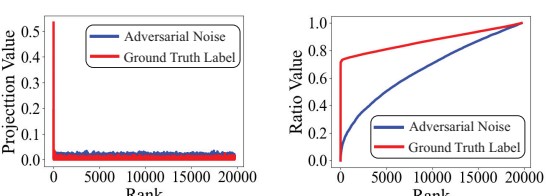

Figure 3: Eigen-projection (left) and concentration ratio (right) of the ground truth label and the adversarial noise on Pubmed. The study is performed for Metattack with a perturbation ratio of $25\%$. By the rank $r = r_0 = 0.2 \min\{N, D'\}$, the signal concentration ratio of the ground truth label is $0.784$. Results on Cora and Citeseer are illustrated in Figure 4 in Section F.1.

$\left\| \mathbf{U}^{(r)\top} \Delta\mathbf{Z} \right\|_F / \|\Delta\mathbf{Z}\|_F$ for $r \in [N]$, which reflects the amount of the adversarial noise projected onto the $r$-th eigenvector of $\mathbf{K}$. We compute the noise concentration ratio for rank $r$ by $\left\| p^{(\Delta\mathbf{Z})}(1:r) \right\|_1$, where $p^{(\Delta\mathbf{Z})}(1:r)$ contains the first $r$ elements of $p^{(\Delta\mathbf{Z})}$. The noise concentration ratio of a rank $r$ reflects the proportion of adversarial noise projected onto the top $r$ eigenvectors of $\mathbf{K}$.

It is observed in Figure 3 and Figure 4 that the eigen-projection of the adversarial noise is relatively uniform over all the eigenvectors. This observation motivates low-rank feature learning on $\mathbf{Z}$, because the low-rank part of the feature matrix $\mathbf{Z}$ covers the dominant information of the ground truth label $\mathbf{Y}$ while learning only a small portion of the adversarial noise. We remark that the low-rank regularization term $\|\mathbf{K}\|_{r_0}$ in (1) is also theoretically motivated by the upper bound for the test loss in the transductive learning setting, presented as (2) in Theorem 3.1. A smaller $\|\mathbf{K}\|_{r_0}$ is obtained by optimizing the training loss in Equation (1), which ensures a smaller kernel complexity (KC) defined in Theorem 3.1, contributing to a smaller generalization bound for transductive node classification.

**Theoretical Justification for the Low-Rank Regularization.** To justify the benefits of applying the low-rank regularization on the latent representations learned by the encoder of the RGAE, we introduce a simple yet standard linear transductive node classification algorithm using the low-rank node representations $\mathbf{Z} \in \mathbb{R}^{N \times d}$ produced by the encoder of the RGAE. We present the generalization bound for the test loss for our low-rank transductive algorithm with the presence of adversarial

noise. Although the latent representations produced by the RGAE encoder are not directly used for downstream node classification, their improved generalization under adversarial perturbations suggests that they capture semantically meaningful and class-discriminative graph structures. Such properties are particularly beneficial for class-conditional generation, where the fidelity of the generated samples depends on the quality and coherence of the latent space.

Define $\mathbf{F}(\mathbf{W}, t) := \mathbf{Z}\mathbf{W}^{(t)}$ as the output of the classifier after the $t$-th iteration of gradient descent on the training loss $L(\mathbf{W}) = \frac{1}{|\mathcal{V}_\mathcal{L}|} \sum_{v_i \in \mathcal{V}_\mathcal{L}} \text{KL}\left(\mathbf{y}_i, [\text{softmax}(\mathbf{Z}\mathbf{W})]_i\right)$ for $t \geq 1$. We have Theorem 3.1, on the Mean Squared Error (MSE) loss of the unlabeled test nodes $\mathcal{V}_\mathcal{U}$ measured by the gap between $[\mathbf{F}(\mathbf{W}, t)]_\mathcal{U}$ and $[\mathbf{Y}]_\mathcal{U}$ when using the low-rank feature $\mathbf{Z}$ with $r_0 \in [N]$, which is the generalization error bound for the linear transductive classifier using $\mathbf{F}(\mathbf{W}) = \mathbf{Z}\mathbf{W}$ to predict the labels of the unlabeled nodes. Following existing works such as (Kothapalli et al., 2023), we employ the MSE loss to provide the generalization error of the GNN-based node classifier. It is remarked that the MSE loss is necessary for the generalization analysis of transductive learning using transductive local Rademacher complexity (Tolstikhin et al., 2014; Yang, 2025). Detailed explanation about Theorem 3.1 is deferred to Section B of the appendix.

**Theorem 3.1.** Let $m \geq cN$ for a constant $c \in (0, 1)$, and $r_0 \in [N]$. Assume that a set $\mathcal{L}$ with $|\mathcal{L}| = m$ is sampled uniformly without replacement from $[N]$, and the remaining nodes $\mathcal{V}_U = \mathcal{V} \setminus \mathcal{V}_L$ are the test nodes. Then for every $x > 0$, with probability at least $1 - \exp(-x)$, after the $t$-th iteration of gradient descent on the training loss $L(\mathbf{W})$ for all $t \geq 1$, we have

$$\mathcal{U}_{\text{test}}(t) := \frac{1}{u} \|[\mathbf{F}(\mathbf{W}, t) - \mathbf{Y}]_\mathcal{U}\|_\text{F}^2 \leq \frac{2c_0 L(\mathbf{K}, \mathbf{Y}, t)}{m} + c_0 \text{KC}(\mathbf{K}) + \frac{c_0 x}{u}, \tag{2}$$

where $c_0$ is a positive number depending on $\mathbf{U}$, $\left\{\widehat{\lambda}_i\right\}_{i=1}^{r_0}$, and $\tau_0$ with $\tau_0^2 = \max_{i \in [N]} \mathbf{K}_{ii}$. $L(\mathbf{K}, \mathbf{Y}, t) := \left\|\left(\mathbf{I}_m - \eta [\mathbf{K}]_{\mathcal{L}, \mathcal{L}}\right)^t [\mathbf{Y}]_\mathcal{L}\right\|_\text{F}^2$, $\mathbf{KC}$ is the kernel complexity of the kernel gram matrix $\mathbf{K} = \mathbf{Z}\mathbf{Z}^\top$ defined by $\text{KC}(\mathbf{K}) = \min_{r_0 \in [N]} r_0 \left(\frac{1}{u} + \frac{1}{m}\right) + \sqrt{\|\mathbf{K}\|_{r_0}} \left(\frac{1}{\sqrt{u}} + \frac{1}{\sqrt{m}}\right)$.

### 3.3 Conditional Generation of Synthetic Graph Structures in Latent Space

To generate synthetic graph structures, consisting of the synthetic nodes and their edges connecting to the original graph, we train a latent diffusion model (LDM) (Rombach et al., 2022) with Classifier-Free Guidance (CFG) (Ho & Salimans, 2022). The trained LDM generates latent features for the synthetic structures, which are then decoded into node features and associated edges. Let $\mathcal{V}_L \subseteq \mathcal{V}$ denote the set of the labeled nodes in the original graph $\mathcal{G}$, and $\mathcal{V}_{\text{syn}}$ denotes the set of the generated synthetic nodes by the RGDM. Let the node attributes of $\mathcal{V}_{\text{syn}}$ generated by the RGDM be $\mathbf{X}_{\text{syn}}$. Let $C$ be the number of classes, and $N_0$ be the number of synthetic nodes for each class. $N' = CN_0 = |\mathcal{V}_{\text{syn}}|$ is the number of the synthetic nodes. Let $Y_{\text{syn}} = \{\widehat{y}_i\}_{i=1}^{N'}$ denote the set of the labels of the synthetic nodes. Let $\widehat{\mathbf{Z}} = \left\{\widehat{\mathbf{Z}}_i\right\}_{i=1}^{N'}$ denote the latent features of the synthetic graph structures. Let $\mathbf{A}_{\text{syn}} \in \mathbb{R}^{N' \times N}$ denote the edges of the synthetic nodes generated by our RGDM. Then the augmented synthetic graph structures are $(\mathcal{V}_{\text{syn}}, \mathbf{X}_{\text{syn}}, \mathbf{A}_{\text{syn}})$. The adjacency matrix of the augmented graph is $\mathbf{A}_{\text{aug}} = [\mathbf{A} \ \mathbf{A}_{\text{syn}}; \mathbf{A}_{\text{syn}} \ \mathbf{A}] \in \mathbb{R}^{(N+N') \times (N+N')}$, and the node features of the augmented graph is $\mathbf{X}_{\text{aug}} = [\mathbf{X}; \mathbf{X}_{\text{syn}}] \in \mathbb{R}^{(N+N') \times D}$. The augmented graph, which is the combination of the original graph $\mathcal{G}$ and the synthetic graph structures, is then denoted by $\mathcal{G}_{\text{aug}} = (\mathcal{V} \cup \mathcal{V}_{\text{syn}}, \mathbf{X}_{\text{aug}}, \mathbf{A}_{\text{aug}})$. The set of labeled nodes in $\mathcal{G}_{\text{aug}}$ is $\mathcal{V}_L \cup \mathcal{V}_{\text{syn}}$ and their labels are $Y_L \cup Y_{\text{syn}}$.

### 3.4 Training GNNs on the Augmented Graph

After obtaining the augmented graph $\mathcal{G}_{\text{aug}}$, we then train either the vanilla GCN (Kipf & Welling, 2017) or the robust GNNs, Pro-GNN (Jin et al., 2020) and SG-GSR (In et al., 2024), on $\mathcal{G}_{\text{aug}}$. Results in Section 4.2 evidence that the GNNs trained on the augmented graph exhibit much better performance and robustness for the semi-supervised node classification task than the models trained on the original attacked graphs perturbed by graph adversarial attacks, including Metattack (Zügner & Günnemann, 2019), Nettack (Zügner et al., 2018), and Topology Attack (Xu et al., 2019).

**Complexity Analysis.** We perform a detailed complexity analysis of the RGDM in Section D of the appendix, demonstrating the improved efficiency of the proposed SHED.

# 4 EXPERIMENTS

We evaluate the performance of the RGDM for augmenting the robustness of the GNNs in the semi-supervised node classification task. Section 4.1 presents the experimental settings for the training of the RGAE and the LDM in the RGDM. Section 4.2 presents the experiment results for node classification under adversarial attacks. In Section 4.3, we conduct an ablation study on the node robustness loss and the edge robustness loss. Additional experiment results are deferred to Section F of the appendix. In Section F.1, we perform the eigen-projection analysis on more datasets. In Section F.2, we compare the performance of RGDM with additional baseline methods. In Section F.3, we propose Frechet Node Distance (FND) and Frechet Edge Distance (FED) to evaluate the quality of the synthetic nodes and edges generated by the RGDM. In Section F.4, we present the training time and the synthetic data generation time of the RGDM. In Section F.5, we present the details of the cross-validation process for selecting the number of synthetic nodes. In Section F.6, we perform experiments for adversarial defense on **large-scale graph datasets**, Reddit (Hamilton et al., 2017) and ogbn-arxiv (Hu et al., 2020). In Section F.7, we perform the sensitivity analysis on the hyperparameters. In Section F.8, we perform experiments on **heterophilic graph datasets**, Chameleon (Pei et al., 2020) and Actor (Tang et al., 2009). The t-SNE visualization of both the real and synthetic node features within the augmented graph is illustrated in Section F.9, We compare RGDM with existing graph augmentation methods based on mix-up and GAN-based SGS generation in Section F.10. The statistical significance analysis of the results in Section 4.2 is performed in Section F.11. In Section F.12, we study the effectiveness of RGDM for purifying adversarially perturbed graphs. In Section F.13, we compare RGDM with diffusion-based graph purification methods. An ablation study on the number of synthetic nodes is conducted in Section F.14. In Section F.15, we compare RGDM with the vanilla DDPM. In Section F.16, we evaluate RGDM under different label ratios. In Section F.17, we compare RGDM with an ablation model that replaces the SHED with the Bi-Level Neighborhood Decoder (BLND) used in DoG (Wang et al., 2025) as the edge decoder.

## 4.1 EXPERIMENTAL SETTINGS

We perform the experimental evaluation on four public graph benchmarks, which are Cora, Citeseer, Pubmed (Sen et al., 2008), and Polblogs (Adamic & Glance, 2005). Details on the statistics of the datasets are deferred to Table 4 in Section E.1 of the appendix. We present the training settings of the RGAE and LDM in RGDM in Section E.2 of the appendix. The number of the synthetic nodes added is selected by cross-validation as described in Section F.5 of the appendix. The starting rank $r_0$ for each dataset under each attack setting in our experiments is selected by performing cross-validation as detailed in Section E.4 of the appendix.

Table 1: Node classification performance (Accuracy±Std) under the non-targeted poisoning attack (Metattack) (Zügner & Günnemann, 2019). The best result is highlighted in bold, and the second-best result is underlined. The improvements of the RGDM over the corresponding baselines are attached in parentheses. These conventions are followed by all the tables in this paper. The comparison with more baseline methods is deferred to Table 5 in Section F.2 of the appendix.

| Dataset | Ptb Rate (%) | GCN | GCORNs | HANG | STABLE | Pro-GNN | SG-GSR | RGDM (GCN) | RGDM (Pro-GNN) | RGDM (SG-GSR) |
|---|---|---|---|---|---|---|---|---|---|---|
| Cora | 0 | 83.5±0.4 | 82.5±0.4 | 80.0±0.3 | 84.7±0.5 | 82.9±0.2 | 85.5±2.3 | 85.2±0.4 (↑ 1.7) | 84.2±0.5 (↑ 1.3) | 85.9±2.1 (↑ 0.3) |
| | 15 | 65.1±0.7 | 72.5±1.2 | 72.8±0.9 | 74.4±0.9 | 76.4±1.2 | 78.5±2.6 | 69.3±0.3 (↑ 4.2) | 78.2±1.1 (↑ 1.8) | 80.0±2.1 (↑ 1.4) |
| | 25 | 47.5±1.9 | 69.4±2.9 | 68.7±1.8 | 69.6±0.6 | 69.7±1.6 | 77.8±2.3 | 61.4±0.7 (↑ 1.8) | 72.3±1.5 (↑ 2.5) | 79.1±1.6 (↑ 1.3) |
| Citeseer | 0 | 71.9±0.5 | 72.6±0.4 | 73.2±0.3 | 74.8±0.5 | 73.2±0.6 | 75.8±2.4 | 73.6±0.8 (↑ 1.7) | 74.4±0.8 (↑ 1.1) | 76.6±1.4 (↑ 0.7) |
| | 15 | 64.5±1.1 | 65.4±2.0 | 70.8±0.8 | 67.5±0.4 | 72.0±1.1 | 73.1±1.4 | 68.7±0.3 (↑ 4.2) | 73.6±0.3 (↑ 1.6) | 76.0±1.1 (↑ 2.8) |
| | 25 | 56.9±2.0 | 65.2±2.2 | 66.4±2.5 | 65.6±1.9 | 68.9±2.7 | 72.7±2.4 | 62.8±0.6 (↑ 5.9) | 70.2±0.6 (↑ 1.2) | 75.3±2.0 (↑ 2.5) |
| Polblogs | 0 | 95.6±0.3 | 95.3±0.8 | 94.7±1.0 | 95.5±0.2 | 93.2±0.6 | 95.9±1.8 | 96.1±0.1 (↑ 0.4) | 95.7±2.2 (↑ 2.5) | 96.1±2.5 (↑ 0.1) |
| | 15 | 64.9±1.9 | 82.0±1.4 | 71.6±1.3 | 87.8±0.3 | 86.0±2.2 | 90.0±2.6 | 69.8±0.3 (↑ 4.8) | 88.0±1.0 (↑ 1.9) | 91.1±2.9 (↑ 1.1) |
| | 25 | 49.2±1.3 | 66.5±2.7 | 65.8±2.3 | 80.0±1.4 | 63.1±2.4 | 87.8±2.1 | 56.4±0.5 (↑ 7.2) | 66.9±1.3 (↑ 3.7) | 88.8±2.8 (↑ 1.0) |
| Pubmed | 0 | 87.1±0.0 | 86.4±0.7 | 85.0±0.2 | 87.7±0.0 | 87.3±0.1 | 87.7±2.7 | 88.1±0.1 (↑ 0.9) | 88.4±0.2 (↑ 1.0) | 90.3±1.5 (↑ 2.5) |
| | 15 | 78.6±0.1 | 76.8±1.8 | 85.0±0.2 | 87.3±0.1 | 87.2±0.0 | 87.3±2.4 | 84.9±0.2 (↑ 6.3) | 88.0±0.2 (↑ 0.8) | 88.7±2.7 (↑ 1.3) |
| | 25 | 75.5±0.1 | 70.7±2.8 | 85.0±0.1 | 86.0±0.1 | 86.7±0.1 | 87.2±2.4 | 81.5±0.1 (↑ 6.0) | 87.5±0.1 (↑ 0.8) | 88.6±2.3 (↑ 1.4) |

Table 2: Node classification performance under targeted attack (Nettack) (Zügner et al., 2018). The comparison with more baseline methods is deferred to Table 6 in Section F.2 of the appendix.

| Dataset | Budget | GCN | GCORNs | HANG | STABLE | Pro-GNN | SG-GSR | RGDM (GCN) | RGDM (Pro-GNN) | RGDM (SG-GSR) |
|---|---|---|---|---|---|---|---|---|---|---|
| Cora | 0 | 81.4±1.0 | 82.6±0.4 | 80.7±1.2 | 85.3±1.8 | 84.7±0.5 | 85.5±2.7 | 83.4±1.0 (↑ 2.0) | 86.0±1.1 (↑ 1.2) | 86.0±1.3 (↑ 0.5) |
| | 3 | 67.9±1.7 | 73.1±0.7 | 73.1±2.8 | 73.8±2.8 | 72.4±0.5 | 74.9±1.2 | 70.4±1.1 (↑ 2.5) | 74.3±1.3 (↑ 1.9) | 76.4±1.5 (↑ 1.5) |
| | 5 | 55.5±1.6 | 66.8±1.0 | 68.8±2.5 | 68.4±1.6 | 66.8±0.7 | 68.0±2.5 | 59.4±1.2 (↑ 3.9) | 69.8±2.0 (↑ 2.9) | 69.8±1.8 (↑ 1.8) |
| Citeseer | 0 | 81.0±1.3 | 81.9±1.1 | 81.0±1.1 | 82.4±1.8 | 82.1±0.8 | 82.8±1.2 | 82.8±1.1 (↑ 1.8) | 83.0±2.0 (↑ 0.9) | 83.7±1.3 (↑ 0.8) |
| | 3 | 63.9±2.6 | 78.1±2.0 | 77.1±2.4 | 80.9±1.5 | 79.6±1.9 | 81.7±1.2 | 68.5±1.1 (↑ 4.5) | 80.8±2.9 (↑ 1.1) | 83.6±2.8 (↑ 1.9) |
| | 5 | 47.6±5.1 | 71.6±2.7 | 73.4±2.4 | 72.5±1.5 | 71.2±2.9 | 73.8±2.0 | 55.7±2.0 (↑ 8.0) | 74.6±6.2 (↑ 3.3) | 75.6±2.4 (↑ 1.8) |
| Polblogs | 0 | 97.0±0.2 | 97.5±0.2 | 97.4±0.5 | 98.0±2.4 | 97.1±0.1 | 97.5±2.3 | 97.6±0.1 (↑ 0.6) | 97.8±0.3 (↑ 0.6) | 98.5±0.2 (↑ 0.9) |
| | 3 | 95.4±0.1 | 95.6±1.0 | 96.6±0.1 | 96.8±1.4 | 96.9±0.1 | 95.4±1.9 | 96.9±0.2 (↑ 1.5) | 97.2±0.5 (↑ 0.2) | 96.4±0.1 (↑ 1.0) |
| | 5 | 93.0±0.4 | 93.4±0.8 | 95.9±0.3 | 96.2±2.8 | 96.1±0.2 | 96.8±2.4 | 94.9±0.4 (↑ 1.9) | 96.8±1.4 (↑ 0.7) | 97.7±0.2 (↑ 0.9) |
| Pubmed | 0 | 88.1±1.3 | 84.5±1.3 | 85.3±1.2 | 90.4±2.5 | 88.4±1.2 | 91.0±1.1 | 89.2±1.3 (↑ 1.0) | 89.5±1.1 (↑ 1.1) | 92.5±1.7 (↑ 1.5) |
| | 3 | 81.2±2.6 | 82.2±1.4 | 84.0±2.1 | 85.6±2.1 | 84.3±2.1 | 86.4±2.2 | 83.3±1.6 (↑ 2.1) | 85.7±1.3 (↑ 1.4) | 89.0±1.0 (↑ 2.6) |
| | 5 | 68.3±5.1 | 72.2±1.1 | 70.5±2.3 | 73.9±1.8 | 72.1±2.3 | 73.5±1.4 | 71.4±1.3 (↑ 3.1) | 76.2±2.0 (↑ 4.0) | 75.7±1.3 (↑ 2.1) |

## 4.2 EXPERIMENTAL RESULTS

We conduct experiments on semi-supervised node classification under the Metattack (Zügner & Günnemann, 2019), the Nettack (Zügner et al., 2018), and the Topology Attack (Xu et al., 2019). The compared methods are detailed in Section E.6 of the appendix. We apply the RGDM on vanilla GCN (Kipf & Welling, 2017) and state-of-the-art robust GNNs, which are Pro-GNN (Jin et al., 2020) and SG-GSR (In et al., 2024). The results for Metattack under the perturbation rates of $15\%$ and $25\%$ are shown in Table 1. Additional results with perturbation rates from $5\%$ to $25\%$ with a step of $5\%$ for the Metattack are deferred to Table 5 in Section F.2 of the appendix. The results for Nettack (Zügner et al., 2018) under attack budgets of 3 and 5 are shown in Table 2. Detailed results for Nettack, with attack budgets varying from 1 to 5 in increments of 1, are deferred to Table 6 in Section F.2 of the appendix. In addition, the results for defending against Topology Attack are deferred to Table 7 in Section F.2 of the appendix. We repeat each experiment 10 times with different seeds for network initialization and report the mean and standard deviation of test accuracy. It is observed from the results that the RGDM models achieve state-of-the-art performance on semi-supervised node classification against adversarial attacks. Benefiting from the edge selection mechanism in RGAE, the RGDM usually improves the performance of the state-of-the-art method by a larger margin under more severe attacks. For example, the RGDM (Pro-GNN) outperforms Pro-GNN by $4.1\%$ on Pubmed under Nettack with the largest attack budget of 5.

## 4.3 ABLATION STUDIES

To validate the effectiveness of the sparsification loss and the low-rank regularization term in Equation (1), we conduct an ablation study on Cora, Citeseer, Polblogs, and Pubmed under the Metattack with the perturbation rate of $25\%$. We evaluate two variants of the RGDM based on the GCN, which are the RGDM without the sparsification loss and the RGDM without the low-rank regularization. We also compare RGDM with existing low-rank adversarial defense methods, (Savostianova et al., 2023; Wu et al., 2022; Entezari et al., 2020; Deng et al., 2022; Zhou et al., 2025; Zhang et al., 2025), with details provided in Section E.5 of the appendix. As shown in Table 3, RGDM outperforms both low-rank baselines across all datasets. Either removing the sparsification loss or the low-rank regularization leads to performance drops, highlighting their effectiveness in enhancing robustness.

Table 3: Ablation study on the sparsification loss and the low-rank regularization term.

| Method | Cora | Citeseer | Polblogs | Pubmed |
|---|---|---|---|---|
| RGDM (GCN) w/o Sparsification Loss ($\mathcal{L}_S^{(t)}$) | 56.72 | 58.97 | 53.24 | 77.92 |
| RGDM (GCN) w/o Low-Rank Regularization ($\|\mathbf{K}\|_{r_0}$) | 60.04 | 60.38 | 54.85 | 80.85 |
| RGDM (GCN, Low-Rank Weight Matrix (Savostianova et al., 2023)) | 59.88 | 61.12 | 55.17 | 80.16 |
| RGDM (GCN, Low-Rank Adjacency Matrix (Entezari et al., 2020)) | 61.01 | 60.25 | 55.03 | 80.02 |
| RGDM (GCN) | **61.42** | **62.84** | **56.47** | **81.52** |

Additional ablation studies are deferred to Section F of the appendix. In Section F.12, we show that RGDM can be adapted to purify the original attacked graph without SGS, and combining the purified original graph with the generated SGS further improves performance. We apply RGDM without purification of the original attacked graph for better efficiency in our experiments. In Section F.13, we compare RGDM with diffusion-based graph purification methods and the results demonstrate that RGDM significantly outperforms existing diffusion-based graph purification methods (He et al., 2025; Luo et al., 2025). In Section F.14, we conduct an ablation study on the number of synthetic nodes. In Section F.15, we compare RGDM to a baseline where the synthetic nodes are generated by DDPM (Ho et al., 2020), showing that synthetic nodes generated by DDPM, which suffer from noise, even hurt the performance of the base model. In Section F.16, we evaluate RGDM under different label ratios and show that it achieves consistent improvements.

## 5 CONCLUSIONS

We present the Robust Graph Diffusion Model (RGDM), marking the first robust graph diffusion model designed to generate synthetic graph structures, which are the synthetic nodes and synthetic edges connecting to them in the graph, for the node classification task. In addition, the RGDM significantly mitigates the negative effects of graph adversarial attacks and generates faithful synthetic graph structures, benefiting from the edge selection mechanism and the low-rank latent feature learning in the RGAE. The experiment results show that incorporating the synthetic graph structures into the original attacked graph for the training of GNNs significantly improves the classification accuracy for the semi-supervised node classification under various types of graph adversarial attacks.

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

## A  THEORETICAL RESULTS

We present the proof of Theorem 3.1 in this section.

**Proof of Theorem 3.1.** It can be verified that at the $t$-th iteration of gradient descent for $t \geq 1$, we have

$$\mathbf{W}^{(t)} = \mathbf{W}^{(t-1)} - \eta \left[\mathbf{Z}\right]_{\mathcal{L}}^{\top} \left[\mathbf{Z}\mathbf{W}^{(t-1)} - \mathbf{Y}\right]_{\mathcal{L}}. \tag{3}$$

It follows by (3) that

$$\left[\mathbf{Z}\right]_{\mathcal{L}} \mathbf{W}^{(t)} = \left[\mathbf{Z}\right]_{\mathcal{L}} \mathbf{W}^{(t-1)} - \eta \mathbf{K}_{\mathcal{L},\mathcal{L}} \left[\mathbf{Z}\mathbf{W}^{(t-1)} - \mathbf{Y}\right]_{\mathcal{L}}, \tag{4}$$

where $\mathbf{K}_{\mathcal{L},\mathcal{L}} \coloneqq \left[\mathbf{Z}\right]_{\mathcal{L}} \left[\mathbf{Z}\right]_{\mathcal{L}}^{\top} \in \mathbb{R}^{m \times m}$. With $\mathbf{F}(\mathbf{W}, t) = \mathbf{Z}\mathbf{W}^{(t)}$, it follows by (4) that

$$\left[\mathbf{F}(\mathbf{W}, t) - \mathbf{Y}\right]_{\mathcal{L}} = \left(\mathbf{I}_m - \eta \left[\mathbf{K}\right]_{\mathcal{L},\mathcal{L}}\right) \left[\mathbf{F}(\mathbf{W}, t-1) - \mathbf{Y}\right]_{\mathcal{L}}.$$

It follows from the above equality and the recursion that

$$\left[\mathbf{F}(\mathbf{W}, t) - \mathbf{Y}\right]_{\mathcal{L}} = -\left(\mathbf{I}_m - \eta \left[\mathbf{K}\right]_{\mathcal{L},\mathcal{L}}\right)^t \left[\mathbf{Y}\right]_{\mathcal{L}}. \tag{5}$$

We apply (Yang, 2025, Corollary 3.7) to obtain the following bound for the test loss $\frac{1}{u}\|[\mathbf{F}(\mathbf{W}, t) - \mathbf{Y}]_{\mathcal{U}}\|_{\mathrm{F}}^2$:

$$\frac{1}{u}\|[\mathbf{F}(\mathbf{W}, t) - \mathbf{Y}]_{\mathcal{U}}\|_{\mathrm{F}}^2 \leq \frac{c_0}{m}\|[\mathbf{F}(\mathbf{W}, t) - \mathbf{Y}]_{\mathcal{L}}\|_{\mathrm{F}}^2 + c_0 \min_{0 \leq Q \leq n} r(u, m, Q) + \frac{c_0 x}{u}, \tag{6}$$

with

$$r(u, m, Q) := Q\left(\frac{1}{u} + \frac{1}{m}\right) + \left(\sqrt{\frac{\sum\limits_{q=Q+1}^{N} \widehat{\lambda}_q}{u}} + \sqrt{\frac{\sum\limits_{q=Q+1}^{N} \widehat{\lambda}_q}{m}}\right),$$

where $c_0$ is a positive constant depending on $\mathbf{U}$, $\left\{\widehat{\lambda}_i\right\}_{i=1}^{r}$, and $\tau_0$ with $\tau_0^2 = \max_{i \in [N]} \mathbf{K}_{ii}$.

It follows from (5) and (6) that for every $r_0 \in [N]$, we have

$$\frac{1}{u}\|[\mathbf{F}(\mathbf{W}, t) - \mathbf{Y}]_{\mathcal{U}}\|_{\mathrm{F}}^2$$

$$\leq \frac{c_0}{m}\left\|\left(\mathbf{I}_m - \eta\,[\mathbf{K}]_{\mathcal{L},\mathcal{L}}\right)^t [\mathbf{Y}]_{\mathcal{L}}\right\|_{\mathrm{F}}^2 + c_0 r_0 \left(\frac{1}{u} + \frac{1}{m}\right) + c_0 \left(\sqrt{\frac{\sum\limits_{q=r_0+1}^{N} \widehat{\lambda}_q}{u}} + \sqrt{\frac{\sum\limits_{q=r_0+1}^{N} \widehat{\lambda}_q}{m}}\right) + \frac{c_0 x}{u}$$

$$\overset{①}{\leq} \frac{2c_0}{m}\left\|\left(\mathbf{I}_m - \eta\,[\mathbf{K}]_{\mathcal{L},\mathcal{L}}\right)^t [\mathbf{Y}]_{\mathcal{L}}\right\|_{\mathrm{F}}^2 + c_0 r_0 \left(\frac{1}{u} + \frac{1}{m}\right) + c_0 \sqrt{\|\mathbf{K}\|_{r_0}} \left(\sqrt{\frac{1}{u}} + \sqrt{\frac{1}{m}}\right) + \frac{c_0 x}{u},$$

(7)

where ① follows from the Cauchy-Schwarz inequality, (5), and $\sum_{q=r_0+1}^{N} \widehat{\lambda}_q = \|\mathbf{K}\|_{r_0}$. (2) then follows directly from (7). □

## B FURTHER EXPLANATION OF THEOREM 3.1

This theorem is proved in Section A of the appendix. It is noted that $\mathcal{U}_{\mathrm{test}}(t)$ is the test loss of the unlabeled nodes measured by the distance between the classifier output $\mathbf{F}(\mathbf{W}, t)$ and $\mathbf{Y}$. There are two terms on the upper bound for the test loss in (2), $L(\mathbf{K}, \mathbf{Y}, t)$ and $\mathrm{KC}(\mathbf{K})$, which are explained as follows. $L(\mathbf{K}, \mathbf{Y}, t)$ corresponds to the training loss of the node classifier with the ground-truth label. $\mathrm{KC}(\mathbf{K})$ is the kernel complexity (KC), which measures the complexity of the kernel gram matrix from the node representation $\mathbf{Z}$ generated by our encoder of the RGAE. We remark that the TNN $\|\mathbf{K}\|_{r_0}$ appears on the RHS of the upper bound (2), theoretically justifying why we learn the low-rank features $\mathbf{K}$ of the RGAE by adding the TNN $\|\mathbf{K}\|_{r_0}$ to the loss of our RGAE. Moreover, when the low frequency property holds, $L(\mathbf{K}, \mathbf{Y}, t)$ would be very small with enough iteration number $t$. $\mathbf{K} = \mathbf{Z}^\top \mathbf{Z}$ is approximately a low-rank matrix of rank $r_0$ since $\mathbf{Z}$ is approximately a rank-$r_0$ matrix with its TNN optimized through the optimization of the encoder of the RGAE. A smaller $\|\mathbf{K}\|_{r_0}$ is obtained by optimizing the training loss in Equation (1), which in turn ensures a smaller kernel complexity (KC) defined in Theorem 3.1, contributing to a smaller generalization bound for transductive node classification.

## C ALGORITHM FOR TRAINING THE RGDM AND GENERATING THE AUGMENTED GRAPH

We present the training algorithm of the RGDM in Algorithm 1, which comprises two steps. The first step, which is from Line 1 to Line 6 in Algorithm 1, describes the training of the RGAE. The second step, which is from Line 7 to Line 13, describes the training of the LDM. At the $t$-th epoch in the training of the RGAE, we use the edge selection mask at the previous epoch, $\mathbf{M}^{(t-1)}$, when updating the parameters $\boldsymbol{\omega}$ of the RGAE. We update $\boldsymbol{\omega}$, including the parameters for predicting the edge selection mask, by performing one step of gradient descent on $\mathcal{L}^{(t)}(\mathbf{M}^{(t-1)})$. After updating the parameters of RGAE in the $t$-th epoch, we compute $\mathbf{M}^{(t)}$, which is used in the next epoch. We initialize $\mathbf{M}^{(0)} = \mathbf{A}$ for the training in the first epoch. Algorithm 2 describes the generation process for the augmented graph $\mathcal{G}_{\mathrm{aug}}$.

---

**Algorithm 1** Training RGDM (Training the GAE and the LDM)

---

**Input:** The input attribute matrix $\mathbf{X}$, the adjacency matrix $\mathbf{A}$, the training epochs of the autoencoder $t_{\text{AE}}$, the labels $Y_L$ of the labeled nodes $\mathcal{V}_L$, and the training epochs of the LDM $t_{\text{LDM}}$

**Output:** The parameters of the RGAE $\boldsymbol{\omega}$ and the parameters of the LDM $\boldsymbol{\theta}$

1: Compute the inter-cluster neighbor map $\mathbf{C}$ and the intra-cluster neighbor map $\mathbf{M}$

2: Initialize the edge selection mask $\mathbf{S}_{\text{inter}}^{(0)} = \mathbf{C}$ and $\mathbf{S}_{\text{intra}}^{(0)} = \mathbf{M}$

3: Initialize the parameter $\boldsymbol{\omega}$ of the RGAE

4: **for** $t \leftarrow 1$ to $t_{\text{AE}}$ **do**

5:     Update $\boldsymbol{\omega}$ by $\boldsymbol{\omega} \leftarrow \boldsymbol{\omega} - \eta \nabla_{\boldsymbol{\omega}} \mathcal{L}_{\text{RGAE}}^{(t)}(\mathbf{S}_{\text{intra}}^{(t-1)}, \mathbf{S}_{\text{inter}}^{(t-1)})$ with $\mathcal{L}_{\text{RGAE}}^{(t)}(\mathbf{S}_{\text{intra}}^{(t-1)}, \mathbf{S}_{\text{inter}}^{(t-1)})$ from Eq.(1)

6:     Compute the edge selection mask $\mathbf{S}_{\text{intra}}^{(t)}$ and $\mathbf{S}_{\text{inter}}^{(t)}$

7: **end for**

8: Initialize the parameter $\boldsymbol{\theta}$ of the LDM.

9: Map the node features $\mathbf{X}$ and the adjacency matrix $\mathbf{A}$ to the latent space using the encoder $g_e$ of the RGAE as $\mathbf{H} = g_e(\mathbf{X}, \mathbf{A})$.

10: **for** $t \leftarrow 1$ to $t_{\text{LDM}}$ **do**

11:     Sample a Gaussian noise $\boldsymbol{\varepsilon} \sim \mathcal{N}(\mathbf{0}, \mathbf{I})$

12:     Get latent feature of $\mathcal{V}_L$ as $\mathbf{H}_L = \{\mathbf{H}_i | v_i \in \mathcal{V}_L\}$

13:     Update $\boldsymbol{\theta}$ by $\boldsymbol{\theta} \leftarrow \boldsymbol{\theta} - \eta \nabla_{\boldsymbol{\theta}} \|\boldsymbol{\varepsilon}_{\theta}(\mathbf{H}_L, Y_L) - \boldsymbol{\varepsilon}\|_2^2$

14: **end for**

15: **return** The parameters of the RGAE $\boldsymbol{\omega}$ and the parameters of the LDM $\boldsymbol{\theta}$

---

**Algorithm 2** Generation of the Augmented Graph $\mathcal{G}_{\text{aug}}$

---

**Input:** The input attribute matrix $\mathbf{X}$, the adjacency matrix $\mathbf{A}$, the training epochs of the GNN $t_{\text{GNN}}$, the number of added nodes $M$, and the labels of the synthetic data $\{\widehat{y}_i\}_{i=1}^{N'}$

**Output:** The augmented graph $\mathcal{G}_{\text{aug}} = (\mathcal{V} \cup \mathcal{V}_{\text{syn}}, \mathbf{X}_{\text{aug}}, \mathbf{A}_{\text{aug}})$.

1: **for** $i \leftarrow 1$ to $M$ **do**

2:     Sample a Gaussian noise $\boldsymbol{\varepsilon} \sim \mathcal{N}(\mathbf{0}, \mathbf{I})$

3:     Set the class label of the $i$-th synthetic node to $\widehat{y}_i$

4:     Generate $\widehat{\mathbf{H}}_i$ from $\boldsymbol{\varepsilon}$ with the LDM for class $\widehat{y}_i$

5: **end for**

6: Decode $\widehat{\mathbf{H}} = \{\widehat{\mathbf{H}}_i\}_{i=1}^{N'}$ to $\mathbf{X}_{\text{syn}}$ and $\mathbf{A}_{\text{syn}}$ with the decoder of the RGAE as $\mathbf{X}_{\text{syn}}, \mathbf{A}_{\text{syn}} = g_d(\widehat{\mathbf{H}})$

7: Get $\mathbf{A}_{\text{aug}} = [\mathbf{A} \ \mathbf{A}_{\text{syn}}; \mathbf{A}_{\text{syn}} \ \mathbf{A}]$ and $\mathbf{X}_{\text{aug}} = [\mathbf{X}; \mathbf{X}_{\text{syn}}]$

8: **return** $\mathcal{G}_{\text{aug}} = (\mathcal{V} \cup \mathcal{V}_{\text{syn}}, \mathbf{X}_{\text{aug}}, \mathbf{A}_{\text{aug}})$

---

# D    DETAILED COMPLEXITY ANALYSIS

In our work, we propose a RGAE with an edge selection mechanism incorporated in the GAT layers for aggregating neighborhood information. In this section, we analyze the inference time complexity and the parameter size of the RGAE with the edge selection mechanism. For comparison, we also analyze the inference time complexity and the parameter size of the GAE that adopts regular GAT layers for aggregating neighborhood information. For ease of comparison, we denote the number of the parameters and the inference cost of all the layers except the GAT layers with the edge selection as $S_{\text{MLP}}$ and $C_{\text{MLP}}$ respectively. Let $E$ denote the number of the number of edges in the graph. Let $L$ denote the number of GAT layers. The inference time complexity of our RGAE is $\mathcal{O}(LDME + ME + C_{\text{MLP}})$, where $\mathcal{O}(LDME)$ is the computation cost of the GAT layers. $\mathcal{O}(ME)$ is the additional complexity for computing the decision masks on all the edges. In comparison, the inference time complexity of the GAE with regular GAT layers is $\mathcal{O}(DLME + C_{\text{MLP}})$. In our experiments, $D = 512$ and $L = 2$. Therefore, the inference time complexity of the RGAE, that is $\mathcal{O}(1025ME + C_{\text{MLP}})$, is comparable to the inference time complexity of GAE that adopts regular GAT layers whose inference time complexity is $\mathcal{O}(1024ME + C_{\text{MLP}})$.

In addition, the parameter size of the RGAE is $L(DM + 2M) + 2M + S_{\text{MLP}}$, where $L(DM + 2M)$ is the number of parameters in the GAT layers and $2M$ is the number of parameters of the linear layer for generating the decision mask. In comparison, the parameter size of the GAE with regular GAT layers is $L(DM + 2M) + S_{\text{MLP}}$. Since $D = 512$ and $L = 2$, the number of parameters of the RGAE, that is $1030M + S_{\text{MLP}}$, is also comparable to the number of parameters in the GAE with regular GAT layers, whose parameters size is $1028M + S_{\text{MLP}}$.

To thoroughly study the complexity of the RGAE, we also analyze its training time complexity. Let $t_{\text{train}}$ be the number of training epochs. The training time complexity of the RGAE is $\mathcal{O}((DLME + ME + C_{\text{MLP}})Nt_{\text{train}} + N^2 t_{\text{train}})$, where $\mathcal{O}((DLME + ME + C_{\text{MLP}})Nt_{\text{train}})$ is the cost for the forward propagation and $\mathcal{O}(N^2 t_{\text{train}})$ is the cost for computing the loss function.

**Complexity Analysis of SHED.** We propose SHED to reconstruct the edges connected to a node in the graph. To show its efficiency, we analyze the inference time complexity and the parameter size of the GAE with SHED. For comparison, we also analyze the inference time complexity and the parameter size of GAE, where SHED is replaced by a regular edge decoder that directly reconstructs the adjacency matrix $\mathbf{A}$. For ease of comparison, we denote the number of parameters and inference cost of all the MLP and GAT layers except SHED as $S_{\text{MLP}}$ and $C_{\text{MLP}}$, respectively. For a node $v_i$ in the graph, let $d_i = \sum_{k=1}^{K} \widehat{\mathbf{C}}_{ik}$ be the number of clusters predicted to be connected to $v_i$. Let $D'$ be the dimension of the input feature for SHED. The inference time complexity of GAE with SHED is $\mathcal{O}(KD' + d_i D'M + C_{\text{MLP}})$, where $\mathcal{O}(KD')$ is the additional complexity for computing the inter-cluster neighbor map and encoding the cluster indices. $\mathcal{O}(d_i D'M)$ is the computation cost for computing the intra-cluster neighbor map. In contrast, the inference time complexity of GAE with a regular edge decoder is $\mathcal{O}(D'KM + C_{\text{MLP}})$. We note that $d_i$ is upper bounded by the degree of the node $v_i$. In most graph datasets, the average degree of nodes is usually very small. For instance, on Pubmed, where the average node degree is 2.25, we have $d_i \leq 2.25$. As a result, $D'(K + d_i M) \ll D'KM$. For example, setting $K = 100$ and $M = 198$ on Pubmed, we find that the inference time complexity of GAE with SHED is $\mathcal{O}(545.5D' + C_{\text{MLP}})$, which is much more efficient than the regular edge decoder whose inference time complexity is $\mathcal{O}(19800D' + C_{\text{MLP}})$. In general, the inference time complexity of GAE with SHED is much lower than that of GAE with a regular edge decoder.

In addition, the parameter size of GAE with SHED is $D'^2 + D'K + D'M + S_{\text{MLP}}$, where $D'^2$ is the number of parameters in the layer for encoding cluster indices. $D'K$ is the number of parameters in the layer for predicting the inter-cluster neighbor map. $D'M$ is the number of parameters in the layer for reconstructing the intra-cluster neighbor map. In contrast, the parameter size of GAE with a regular edge decoder is $D'KM + S_{\text{MLP}}$. Because $K + M \ll KM$, the parameter size of GAE with SHED is much smaller than that of GAE with a regular edge decoder. For example, $KM = N = 19717$ on Pubmed with $M = 198$ and $K = 100$.

# E  ADDITIONAL EXPERIMENTAL SETTINGS

## E.1  DATASETS

Following previous works on adversarial attacks and defense of GNNs (Jin et al., 2020; Zügner & Günnemann, 2019; Entezari et al., 2020), we evaluate the RGDM on four public benchmark datasets for node classification, including three citation graphs, which are Cora, Citeseer, and Pubmed, and one blog graph, that is, Polblogs. Following previous works on graph adversarial attacks, we evaluate our method and baselines on the largest connected component (LCC) of the graphs. We show the statistics of the datasets in Table 4.

Table 4: Statistics of Cora, Citeseer, Polblogs, and Pubmed.

| Datasets | # Node | # Edge | Classes | Features |
|----------|--------|--------|---------|----------|
| Cora | 2,485 | 5,069 | 7 | 1,433 |
| Citeseer | 2,110 | 3,668 | 6 | 3,703 |
| Polblogs | 1,222 | 16,714 | 2 | 1,222 |
| Pubmed | 19,717 | 44,338 | 3 | 500 |
| ogbn-arxiv | 169,343 | 1,166,243 | 40 | 128 |
| Reddit | 232,965 | 11,606,919 | 41 | 602 |

## E.2  TRAINING SETTINGS

The training of the RGAE is divided into two phases. In the first phase, we pre-train the RGAE by only optimizing the consistency loss $\mathcal{L}_C^{(t)}$ in Equation (1) for 2000 epochs. In the second phase, we optimize $\mathcal{L}^{(t)}$ for another 1000 epochs. We use the Adam optimizer with a learning rate of $0.001$ for the training. The weight decay is set to $1 \times 10^{-5}$. We train the LDM in the RGDM after finishing the training of the RGAE. We use the Adam optimizer with a learning rate of $0.0002$ to train the LDM for 3000 epochs.

## E.3  ATTACK SETTINGS

**Non-targeted Adversarial Attacks (Metattack) (Zügner & Günnemann, 2019).** We first evaluate the robustness of our method against the non-targeted adversarial attack method Metattack. Metattack treats the graph as a hyperparameter to optimize and uses the meta-gradients to solve the bi-level optimization problem, which minimizes the node classification accuracy. We follow the implementation in (Zügner & Günnemann, 2019). As Metattack has several variants, we follow (Jin et al., 2020) and adopt the most destructive attack version, Meta-Self, on Cora, Citeseer, and Polblogs datasets. On Pubmed, we adopt the approximate version of Meta-Self, A-Meta-Self, following the settings in (Jin et al., 2020). We measure the strength of the attack by the perturbation rate, which is the ratio of perturbed edges among all the edges in the graph. We evaluate the RGDM and all the baselines with the perturbation rates ranging from 0 to $25\%$ with a step of $5\%$.

**Targeted Adversarial Attack (Nettack) (Zügner et al., 2018).** We adopt Nettack as the targeted attack method in evaluating the robustness of our method. Nettack manipulates the edges and node features to degrade the classification accuracy on the target nodes while minimizing the change in the graph's degree distribution and the feature co-occurrences. We use the default attack settings in the original implementation in (Zügner et al., 2018). The nodes in the test set whose degree is larger than 10 are set as the target nodes. The number of perturbations made on every targeted node is defined as the attack budget. Following (Jin et al., 2020), we evaluate the RGDM and all the baselines with attack budgets ranging from 1 to 5 with a step size of 1. Following the settings in (Jin et al., 2020), we only sample $10\%$ of the target nodes for the evaluation on the Nettack.

**Gradient-based Attack Method (Topology Attack) (Xu et al., 2019).** We evaluate the robustness of the RGDM against a gradient-based attack method named Topology Attack (Xu et al., 2019). The topology Attack proposes a projected gradient descent (PGD) adversarial attack method that perturbs the edges in a graph to degrade the overall node classification accuracy on the perturbed graph. We adopt the same settings in (Xu et al., 2019) to attack the graphs in our experiments.

Similar to the settings for Metattack, we measure the strength of the attack by the perturbation rate, which is the ratio of the perturbed edges among all the edges in the graph. We evaluate the RGDM and all the baselines with perturbation rates ranging from $0$ to $25\%$ with a step of $5\%$.

### E.4 Tuning the Staring Rank $r_0$ by Cross-Validation

We tune the rank $r_0$ for the TNN, $\|\mathbf{K}\|_{r_0}$, in Equation (1) by standard cross-validation on each dataset. Let $r_0 = \lceil \gamma \min\{N, d\} \rceil$ where $\gamma$ is the rank ratio. We select the values of $\gamma$ by performing 5-fold cross-validation on $20\%$ of the training data in each dataset. The value of $\gamma$ is selected from $\{0.1, 0.2, 0.3, 0.4, 0.5, 0.6, 0.7, 0.8, 0.9\}$.

### E.5 Details on Learning the Low-Rank Weight Matrix and Low-Rank Adjacency Matrix

To evaluate the robustness of the proposed low-rank regularization in RGDM, we construct two ablation baselines by incorporating existing low-rank learning techniques into the RGAE. Next, we detail the implementation of each ablation model. Following the setting in (Savostianova et al., 2023), we regularize the weight matrices of the encoder in the RGAE to approximate orthonormality by adding a penalty term to the training loss of robust RAE, that is $\mathcal{L}_{\text{orth}} = \sum_l \left\| \mathbf{W}_l^\top \mathbf{W}_l - \mathbf{I} \right\|_F^2$, where $\mathbf{W}_l$ denotes the weight matrix of the $l$-th layer and $\mathbf{I}$ is the identity matrix. This regularization encourages each weight matrix to be close to orthonormal, promoting low effective rank and spectral concentration. The strength of this regularization is controlled by the scale factor for $\mathcal{L}_{\text{orth}}$, which is tuned via cross-validation following the setting in (Savostianova et al., 2023). For the ablation model that learns the low-rank adjacency matrix, we replace the original attacked adjacency matrix $\mathbf{A}$ with its low-rank approximation. Following (Entezari et al., 2020), we compute the top-$r_A$ eigenvectors $\mathbf{U}_{r_A}$ of the symmetrically normalized adjacency matrix $\tilde{\mathbf{A}} = \mathbf{D}^{-1/2}\mathbf{A}\mathbf{D}^{-1/2}$, and reconstruct the low-rank adjacency matrix as $\mathbf{A}_{\text{LR}} = \mathbf{D}^{1/2}\mathbf{U}_{r_A}\Lambda_{r_A}\mathbf{U}_{r_A}^\top\mathbf{D}^{1/2}$, where $\Lambda_{r_A}$ is the diagonal matrix containing the top-$r_A$ eigenvalues. The rank $r_A$ is selected via cross-validation following (Entezari et al., 2020). For all datasets, we select $r_A$ from the same candidate sets as described in Section E.4.

### E.6 Compared Methods

In our empirical evaluation, we comprehensively benchmark the proposed RGDM against various GNNs, including models that exemplify advances in expressive power, structural modeling, and resilience to adversarial or noisy perturbations. The standard GNN baselines include Graph Convolutional Networks (GCN) (Kipf & Welling, 2017), Graph Attention Networks (GAT) (Veličković et al., 2018), and Relational GCNs (RGCN) (Zhu et al., 2019), which serve as foundational architectures widely adopted for semi-supervised node classification. Additionally, we incorporate GCN-SVD (Entezari et al., 2020), which leverages truncated singular value decomposition to mitigate the impact of noisy graph spectra. To rigorously assess the robustness of RGDM under adversarial settings, we further compare it with the state-of-the-art robust GNNs explicitly designed to enhance stability and generalization in the presence of graph corruption. These include Graph-Bel (Song et al., 2022), which integrates belief propagation with GNNs for noise-tolerant inference; UAG (Feng et al., 2021), which unifies adversarial training with graph structure learning; GCORNs (Abbahaddou et al., 2024), a framework that co-regularizes neighborhood consistency and robustness objectives; GADC (Liu et al., 2024), which employs adaptive denoising through graph diffusion mechanisms; and HANG (Zhao et al., 2023), a hierarchical adversarial defense method. We also include Pro-GNN (Jin et al., 2020), which jointly optimizes graph structure and model parameters via low-rank and sparsity constraints, and STABLE (Li et al., 2022b), which stabilizes message passing through spectral smoothing.

# F ADDITIONAL EXPERIMENTAL RESULTS

## F.1 EIGEN-PROJECTION AND CONCENTRATION RATIO OF THE GROUND TRUTH LABEL AND THE ADVERSARIAL NOISE

The eigen-projection and the concentration ratio of the ground truth label and the adversarial noise Cora, Citeseer, and Pubmed are illustrated in Figure 4.

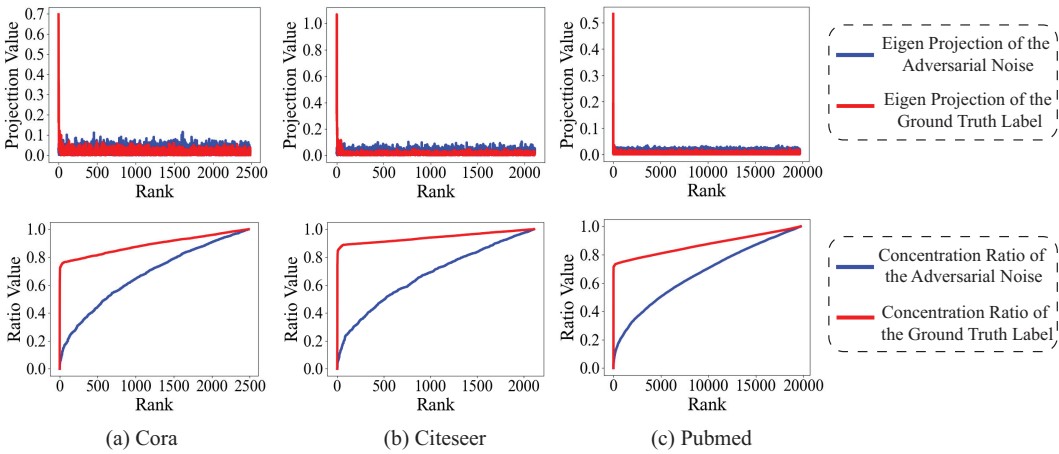

(a) Cora      (b) Citeseer      (c) Pubmed

Figure 4: Eigen-projection (first row) and concentration ratio (second row) of the ground truth label and the adversarial noise on Cora, Citeseer, and Pubmed. The study in this figure is performed for Metattack with a perturbation ratio of 25%. By the rank $r = r_0 = 0.2 \min\{N, D'\}$, the signal concentration ratio of the ground truth label for Cora, Citeseer, and Pubmed are 0.844, 0.809, and 0.784 respectively.

## F.2 DETAILED SEMI-SUPERVISED NODE CLASSIFICATION RESULTS

The detailed experiment results for the RGDM under the Metattack, Nettack, and Topology Attack are shown in Table 5, Table 6 and Table 7.

Table 5: Node classification performance (Accuracy±Std) under the non-targeted poisoning attack (Metattack) (Zügner & Günnemann, 2019). The improvements of the RGDM over the corresponding baselines are attached in parentheses after the results of the RGDM.

| Dataset | Ptb Rate (%) | GCN | GAT | RGCN | GCN-SVD | GraphBel | UAG | GCORNs | GADC | HANG | Pro-GNN | RGDM (GCN) | RGDM (HANG) | RGDM (Pro-GNN) |
|---|---|---|---|---|---|---|---|---|---|---|---|---|---|---|
| Cora | 0 | 83.50±0.44 | 83.97±0.65 | 83.09±0.44 | 80.63±0.45 | 83.42±0.52 | 82.05±0.51 | 82.56±0.48 | 83.22±0.61 | 80.07±0.32 | 82.98±0.23 | 85.23±0.44 (↑ 1.73) | 82.31±0.51 (↑ 2.24) | 84.29±0.52 (↑ 1.31) |
| | 5 | 76.55±0.79 | 80.44±0.74 | 77.42±0.39 | 78.39±0.54 | 82.78±0.39 | 79.13±0.59 | 75.91±1.88 | 75.68±1.10 | 78.12±0.72 | 82.27±0.45 | 80.19±0.39 (↑ 3.64) | 81.04±0.59 (↑ 2.92) | 83.19±0.39 (↑ 0.92) |
| | 10 | 70.39±1.28 | 75.61±0.59 | 72.22±0.38 | 71.47±0.83 | 77.91±0.86 | 75.16±0.76 | 74.63±2.02 | 73.70±1.17 | 75.96±0.62 | 79.03±0.59 | 76.73±0.38 (↑ 6.34) | 78.03±0.76 (↑ 2.07) | 81.16±0.86 (↑ 2.13) |
| | 15 | 65.10±0.71 | 69.78±1.28 | 66.82±0.39 | 66.69±1.18 | 76.01±1.12 | 71.03±0.64 | 72.57±1.24 | 71.39±1.22 | 72.89±0.96 | 76.40±1.27 | 69.37±0.39 (↑ 4.27) | 76.39±0.64 (↑ 3.50) | 78.29±1.12 (↑ 1.89) |
| | 20 | 59.56±2.72 | 59.94±0.92 | 59.27±0.37 | 58.94±1.13 | 68.78±5.84 | 65.71±0.89 | 71.94±1.51 | 71.20±1.49 | 70.36±1.09 | 73.32±1.56 | 65.97±0.37 (↑ 6.41) | 73.25±0.89 (↑ 2.89) | 74.85±1.84 (↑ 1.53) |
| | 25 | 47.53±1.91 | 54.78±0.74 | 50.51±0.78 | 52.06±1.19 | 56.54±2.58 | 60.82±1.08 | 69.48±2.94 | 68.29±1.36 | 68.72±1.87 | 69.72±1.69 | 61.42±0.78 (↑ 1.89) | 70.44±1.08 (↑ 1.72) | 72.31±1.58 (↑ 2.59) |
| Citeseer | 0 | 71.96±0.55 | 73.26±0.83 | 71.20±0.83 | 70.65±0.32 | 72.26±0.83 | 72.10±0.63 | 72.63±0.40 | 73.05±0.30 | 73.26±0.38 | 73.28±0.69 | 73.66±0.83 (↑ 1.70) | 74.21±0.63 (↑ 0.95) | 74.47±0.83 (↑ 1.19) |
| | 5 | 70.88±0.62 | 72.89±0.83 | 70.50±0.43 | 68.84±0.72 | 71.31±0.43 | 70.51±0.97 | 68.18±1.68 | 68.67±0.89 | 73.09±0.34 | 72.93±0.57 | 73.14±0.43 (↑ 2.26) | 74.04±0.97 (↑ 0.95) | 74.29±0.43 (↑ 1.36) |
| | 10 | 67.55±0.89 | 70.63±0.48 | 67.71±0.30 | 68.87±0.62 | 71.63±0.30 | 69.54±0.56 | 66.24±1.52 | 68.66±1.48 | 72.43±0.52 | 72.51±0.75 | 71.35±0.30 (↑ 3.80) | 73.85±0.56 (↑ 1.42) | 74.03±0.30 (↑ 1.52) |
| | 15 | 64.52±1.11 | 69.02±1.09 | 65.69±0.37 | 63.26±0.96 | 69.82±0.37 | 65.95±0.94 | 65.48±2.03 | 68.29±1.88 | 70.82±0.87 | 72.03±1.11 | 68.72±0.37 (↑ 4.20) | 73.21±0.94 (↑ 2.39) | 73.65±0.37 (↑ 1.62) |
| | 20 | 62.03±2.49 | 61.04±1.52 | 62.49±1.22 | 58.55±1.09 | 64.22±1.22 | 59.30±1.40 | 65.55±1.55 | 67.87±2.57 | 66.19±2.38 | 70.02±2.28 | 66.17±1.22 (↑ 4.14) | 69.57±1.40 (↑ 3.38) | 71.87±1.22 (↑ 1.85) |
| | 25 | 56.94±2.09 | 61.85±1.12 | 55.35±0.66 | 57.18±1.87 | 66.23±0.66 | 59.89±1.47 | 65.23±2.26 | 64.82±2.31 | 66.40±2.57 | 68.95±2.78 | 62.84±0.66 (↑ 5.90) | 69.05±1.47 (↑ 2.65) | 70.24±0.66 (↑ 1.29) |
| Polblogs | 0 | 95.69±0.38 | 95.35±0.20 | 95.22±0.14 | 95.31±0.18 | 85.13±2.22 | 90.13±2.22 | 95.31±0.88 | 95.54±0.17 | 94.77±1.07 | 93.20±0.64 | 96.13±0.14 (↑ 0.44) | 94.95±2.22 (↑ 0.18) | 95.70±2.22 (↑ 2.50) |
| | 5 | 73.07±0.80 | 83.69±1.45 | 74.34±0.19 | 89.09±0.22 | 51.84±2.38 | 84.84±2.38 | 89.32±0.33 | 90.52±0.27 | 80.19±2.52 | 93.29±0.18 | 78.57±0.19 (↑ 5.50) | 88.25±2.38 (↑ 8.06) | 94.33±1.38 (↑ 1.04) |
| | 10 | 70.72±1.13 | 76.32±0.85 | 71.04±0.34 | 81.24±0.49 | 56.54±2.30 | 75.54±2.30 | 87.34±1.77 | 86.30±1.45 | 74.92±2.32 | 89.42±1.09 | 74.33±0.34 (↑ 3.61) | 80.47±2.30 (↑ 5.55) | 91.35±1.30 (↑ 1.93) |
| | 15 | 64.96±1.91 | 68.80±1.14 | 67.28±0.38 | 68.10±2.73 | 53.41±2.08 | 66.41±2.08 | 82.07±1.41 | 83.45±1.75 | 71.65±1.34 | 86.04±2.21 | 69.85±0.38 (↑ 4.89) | 78.49±2.08 (↑ 6.84) | 88.01±1.08 (↑ 1.97) |
| | 20 | 51.27±1.23 | 51.50±1.63 | 59.89±0.34 | 57.33±2.15 | 52.18±0.54 | 58.18±0.54 | 69.92±1.26 | 72.84±1.24 | 66.27±2.39 | 79.56±5.68 | 59.38±0.34 (↑ 8.11) | 72.51±0.54 (↑ 6.24) | 81.34±0.54 (↑ 1.78) |
| | 25 | 49.23±1.36 | 51.19±1.49 | 56.02±0.56 | 48.66±9.93 | 51.39±1.36 | 55.39±1.36 | 66.59±2.74 | 65.52±1.93 | 65.80±2.33 | 63.18±2.40 | 56.47±0.56 (↑ 7.24) | 70.79±1.36 (↑ 4.99) | 66.92±1.36 (↑ 3.74) |
| Pubmed | 0 | 87.19±0.09 | 83.73±0.40 | 86.16±0.18 | 83.44±0.21 | 84.02±0.26 | 87.06±0.06 | 86.42±0.71 | 87.26±0.25 | 85.08±0.20 | 87.33±0.18 | 88.10±0.18 (↑ 0.91) | 87.15±0.11 (↑ 2.07) | 88.41±0.26 (↑ 1.08) |
| | 5 | 83.09±0.13 | 78.00±0.44 | 81.08±0.20 | 83.41±0.15 | 83.91±0.26 | 86.39±0.06 | 82.96±0.89 | 81.64±0.64 | 85.08±0.18 | 87.25±0.09 | 86.99±0.20 (↑ 3.90) | 87.02±0.11 (↑ 1.94) | 88.26±0.26 (↑ 1.01) |
| | 10 | 81.21±0.09 | 74.93±0.38 | 77.51±0.27 | 83.27±0.21 | 84.62±0.26 | 85.70±0.07 | 80.76±1.65 | 80.43±1.64 | 85.17±0.23 | 87.25±0.09 | 85.83±0.27 (↑ 4.62) | 87.13±0.14 (↑ 1.96) | 88.15±0.26 (↑ 0.90) |
| | 15 | 78.66±0.12 | 71.13±0.51 | 73.91±0.25 | 83.10±0.18 | 84.83±0.20 | 84.76±0.08 | 76.82±1.83 | 75.72±1.24 | 85.00±0.22 | 87.20±0.09 | 84.97±0.25 (↑ 6.31) | 88.09±0.20 (↑ 0.89) | |
| | 20 | 77.35±0.19 | 68.21±0.96 | 71.18±0.31 | 83.01±0.22 | 84.89±0.45 | 83.88±0.05 | 71.28±1.52 | 72.80±1.89 | 85.20±0.19 | 87.15±0.15 | 82.45±0.31 (↑ 5.10) | 86.13±0.11 (↑ 0.93) | 88.00±0.45 (↑ 0.85) |
| | 25 | 75.50±0.17 | 65.41±0.77 | 67.95±0.15 | 82.72±0.18 | 85.07±0.15 | 83.66±0.06 | 70.75±2.86 | 71.60±2.50 | 85.06±0.17 | 86.76±0.19 | 81.52±0.15 (↑ 6.02) | 86.26±0.12 (↑ 1.20) | 87.58±0.15 (↑ 0.82) |

## F.3 QUALITY EVALUATION OF THE AUGMENTED GRAPH

This paper introduces a novel graph diffusion model named RGDM, which synthesizes the graph structures. The synthetic graph structures generated by the RGDM are subsequently combined with the original attacked graph to form an augmented graph. By incorporating an edge selection mechanism in the RGDM, we aim to improve the robustness of the GNNs trained on the augmented graphs against the graph adversarial attacks. In Section 4.2, we have shown that both the vanilla GCN and

Table 6: Node classification performance (Accuracy±Std) under targeted attack (Nettack) (Zügner et al., 2018). The improvements of RGDM over the corresponding baselines are attached in parentheses after the results of the RGDM.

| Dataset | Attack Budget | GCN | GAT | RGCN | GCN-SVD | GraphBel | UAG | GCORNs | GADC | HANG | Pro-GNN | RGDM (GCN) | RGDM (HANG) | RGDM (Pro-GNN) |
|---|---|---|---|---|---|---|---|---|---|---|---|---|---|---|
| Cora | 0 | 81.43±1.04 | 82.14±1.13 | 81.34±1.55 | 81.10±1.34 | 71.55±2.01 | 82.34±1.08 | 82.61±0.41 | 81.36±1.83 | 80.75±1.27 | 84.78±0.59 | 83.43±1.08 (↑ 2.00) | 82.10±1.34 (↑ 1.35) | 86.00±1.13 (↑ 1.22) |
|  | 1 | 75.06±1.02 | 76.04±2.08 | 76.75±1.71 | 77.23±1.82 | 63.73±2.25 | 81.75±1.16 | 81.05±1.14 | 80.05±1.53 | 76.99±2.16 | 83.85±0.51 | 78.41±1.16 (↑ 3.35) | 80.39±1.82 (↑ 3.40) | 84.79±2.08 (↑ 0.94) |
|  | 2 | 70.60±1.10 | 70.24±1.43 | 70.96±1.14 | 72.53±1.60 | 62.41±2.94 | 77.96±1.84 | 78.47±1.32 | 75.90±2.29 | 76.51±2.60 | 78.65±0.66 | 73.79±1.84 (↑ 3.19) | 79.14±1.60 (↑ 2.63) | 80.02±1.43 (↑ 1.37) |
|  | 3 | 67.95±1.72 | 65.54±1.34 | 66.51±1.60 | 66.75±1.54 | 61.20±2.08 | 72.51±1.14 | 73.17±0.74 | 71.94±1.45 | 73.13±2.85 | 72.40±0.53 | 70.46±1.14 (↑ 2.51) | 76.31±1.54 (↑ 3.18) | 74.31±1.34 (↑ 1.91) |
|  | 4 | 61.57±1.47 | 61.69±0.90 | 59.28±2.68 | 60.72±1.63 | 56.51±2.72 | 70.28±2.52 | 71.26±0.62 | 70.93±2.19 | 72.53±2.14 | 70.12±0.66 | 64.37±2.52 (↑ 2.80) | 75.12±1.63 (↑ 2.59) | 72.79±0.90 (↑ 2.67) |
|  | 5 | 55.54±1.66 | 58.31±2.03 | 55.30±1.66 | 57.71±1.82 | 51.93±2.77 | 66.30±1.21 | 66.89±1.08 | 64.49±2.42 | 68.80±2.55 | 66.89±0.73 | 59.44±1.21 (↑ 3.90) | 69.81±1.82 (↑ 1.01) | 69.81±2.03 (↑ 2.92) |
| Citeseer | 0 | 81.02±1.35 | 82.26±2.04 | 80.25±1.12 | 81.08±1.34 | 76.55±2.23 | 81.25±1.12 | 81.94±1.17 | 80.63±1.04 | 81.09±1.12 | 82.12±0.81 | 82.87±1.12 (↑ 1.85) | 82.84±1.34 (↑ 1.75) | 83.05±2.04 (↑ 0.93) |
|  | 1 | 78.41±1.62 | 81.27±1.38 | 78.25±0.73 | 80.16±2.04 | 68.89±2.67 | 80.25±0.73 | 80.79±0.89 | 79.02±2.84 | 79.05±1.38 | 81.75±0.79 | 80.39±0.73 (↑ 1.98) | 81.40±2.04 (↑ 2.35) | 82.47±1.38 (↑ 0.72) |
|  | 2 | 74.92±2.54 | 77.43±2.89 | 75.40±2.04 | 79.84±0.73 | 67.62±2.11 | 79.40±2.04 | 79.65±2.84 | 78.22±1.29 | 77.94±2.29 | 81.27±0.95 | 77.15±2.04 (↑ 2.23) | 80.11±0.73 (↑ 2.17) | 82.30±2.89 (↑ 1.03) |
|  | 3 | 63.97±2.69 | 60.85±2.99 | 60.31±1.19 | 77.14±2.86 | 60.63±2.87 | 78.31±1.19 | 78.10±2.06 | 78.11±1.39 | 77.14±2.48 | 79.68±1.98 | 68.53±1.19 (↑ 4.56) | 78.95±2.86 (↑ 1.81) | 80.85±2.99 (↑ 1.17) |
|  | 4 | 55.40±2.60 | 61.59±2.64 | 55.49±1.75 | 69.52±2.31 | 53.17±6.48 | 77.49±1.75 | 77.75±1.66 | 77.49±1.54 | 78.41±1.62 | 77.78±2.84 | 59.57±1.75 (↑ 4.17) | 78.99±1.31 (↑ 0.58) | 79.11±2.64 (↑ 1.33) |
|  | 5 | 47.62±5.17 | 55.56±6.28 | 47.44±2.01 | 69.21±2.48 | 48.73±2.60 | 71.44±2.01 | 71.66±2.79 | 70.54±1.57 | 73.49±2.48 | 71.27±2.99 | 55.71±2.01 (↑ 8.09) | 75.92±2.48 (↑ 2.43) | 74.60±6.28 (↑ 3.33) |
| Polblogs | 0 | 97.03±0.24 | 97.25±0.32 | 97.04±0.11 | 97.50±0.22 | 71.35±2.01 | 97.11±0.11 | 97.52±0.26 | 97.38±0.34 | 97.41±0.50 | 97.12±0.15 | 97.69±0.11 (↑ 0.66) | 97.65±0.22 (↑ 0.24) | 97.80±0.32 (↑ 0.68) |
|  | 1 | 96.83±0.17 | 97.22±0.25 | 97.00±0.07 | 97.56±0.30 | 68.17±2.25 | 96.00±0.07 | 97.61±0.59 | 97.18±0.35 | 97.37±0.37 | 96.83±0.06 | 97.67±0.07 (↑ 0.84) | 97.63±0.20 (↑ 0.26) | 97.69±0.25 (↑ 0.86) |
|  | 2 | 95.61±0.20 | 96.11±0.65 | 95.87±0.23 | 97.12±0.09 | 65.48±2.85 | 97.02±0.23 | 96.08±0.45 | 95.37±0.59 | 96.89±0.16 | 97.17±0.12 | 97.10±0.23 (↑ 1.49) | 97.24±0.09 (↑ 0.35) | 97.58±0.65 (↑ 0.41) |
|  | 3 | 95.41±0.18 | 95.81±0.56 | 95.59±0.27 | 96.61±0.14 | 62.59±1.99 | 96.59±0.27 | 95.65±1.01 | 95.47±0.13 | 96.65±0.15 | 96.93±0.12 | 96.99±0.27 (↑ 1.58) | 97.01±0.14 (↑ 0.36) | 97.22±0.56 (↑ 0.29) |
|  | 4 | 94.24±0.24 | 94.80±0.66 | 94.37±0.26 | 96.17±0.19 | 58.68±0.40 | 96.07±0.26 | 94.44±0.53 | 94.97±0.61 | 96.26±0.53 | 96.89±0.16 | 95.44±0.26 (↑ 1.20) | 97.03±0.19 (↑ 0.77) | 97.12±0.66 (↑ 0.23) |
|  | 5 | 93.00±0.48 | 93.28±1.43 | 93.20±0.43 | 95.13±0.25 | 59.02±2.19 | 95.60±0.43 | 93.42±0.85 | 93.27±0.46 | 95.91±0.33 | 96.13±0.25 | 94.90±0.43 (↑ 1.90) | 96.70±0.25 (↑ 0.79) | 96.88±1.43 (↑ 0.75) |
| Pubmed | 0 | 88.13±1.35 | 87.03±1.13 | 84.87±1.35 | 87.27±1.74 | 85.47±1.45 | 87.31±1.35 | 84.56±1.39 | 85.49±1.16 | 85.35±1.20 | 88.46±1.20 | 89.21±1.35 (↑ 1.09) | 87.28±1.74 (↑ 1.93) | 89.58±1.13 (↑ 1.12) |
|  | 1 | 87.04±1.62 | 84.95±2.08 | 83.75±1.71 | 84.15±2.25 | 84.34±2.33 | 86.57±1.71 | 83.88±1.94 | 84.58±1.28 | 84.42±2.01 | 87.58±2.01 | 88.03±1.71 (↑ 0.99) | 86.55±1.94 (↑ 1.22) | 88.80±2.08 (↑ 1.22) |
|  | 2 | 84.11±2.54 | 83.90±1.43 | 84.21±1.14 | 83.90±2.94 | 84.27±2.25 | 85.21±1.14 | 84.40±1.73 | 84.17±1.53 | 84.38±2.05 | 85.87±2.05 | 85.33±1.14 (↑ 2.21) | 86.35±1.94 (↑ 1.97) | 87.14±1.43 (↑ 1.27) |
|  | 3 | 81.25±2.69 | 81.27±1.34 | 82.48±1.60 | 81.22±2.08 | 84.11±2.67 | 83.88±1.60 | 82.26±1.49 | 84.35±1.33 | 84.01±2.12 | 84.35±2.12 | 83.77±1.60 (↑ 2.12) | 85.21±1.08 (↑ 1.20) | 85.79±1.34 (↑ 1.44) |
|  | 4 | 76.35±2.60 | 78.54±0.90 | 80.02±2.68 | 77.02±2.72 | 79.03±2.22 | 80.00±2.32 | 81.06±2.72 | 78.01±2.20 | 79.73±2.31 | 80.04±2.31 | 78.45±2.32 (↑ 2.10) | 81.65±2.23 (↑ 1.92) | 81.95±0.90 (↑ 1.91) |
|  | 5 | 68.32±5.17 | 74.25±2.03 | 73.14±2.66 | 71.84±2.77 | 70.11±2.36 | 72.20±1.39 | 72.28±1.13 | 71.26±1.61 | 70.59±2.31 | 72.18±2.31 | 71.42±1.39 (↑ 3.10) | 74.43±1.34 (↑ 3.84) | 76.27±2.03 (↑ 4.09) |

Table 7: Node classification performance (Accuracy±Std) under Topology Attack (Xu et al., 2019). The improvements of the RGDM over the corresponding baselines are attached in parentheses after the results of the RGDM.

| Dataset | Ptb Rate (%) | GCN | GAT | RGCN | GCN-SVD | GraphBel | UAG | GCORNs | GADC | HANG | Pro-GNN | RGDM(GCN) | RGDM(HANG) | RGDM(Pro-GNN) |
|---|---|---|---|---|---|---|---|---|---|---|---|---|---|---|
| Cora | 0 | 83.50±0.44 | 83.97±0.65 | 83.09±0.44 | 80.63±0.45 | 83.42±0.52 | 82.05±0.51 | 82.63±0.47 | 83.21±0.72 | 80.07±0.32 | 82.98±0.23 | 85.23±0.44 (↑ 1.73) | 82.31±0.51 (↑ 2.24) | 84.29±0.52 (↑ 1.31) |
|  | 5 | 75.50±0.44 | 76.97±0.65 | 75.04±1.32 | 75.82±1.24 | 77.01±1.33 | 76.22±1.24 | 75.96±1.81 | 76.83±1.25 | 76.99±1.52 | 77.63±1.85 | 77.92±2.08 (↑ 2.42) | 79.39±1.33 (↑ 2.40) | 81.10±1.16 (↑ 3.47) |
|  | 10 | 72.02±1.32 | 74.10±1.84 | 73.05±1.25 | 74.00±1.76 | 74.00±1.57 | 76.00±1.76 | 74.67±2.03 | 75.90±1.84 | 76.99±1.52 | 77.63±1.85 | 74.22±1.43 (↑ 2.20) | 79.44±1.57 (↑ 2.45) | 80.36±1.84 (↑ 2.73) |
|  | 15 | 69.42±1.71 | 70.55±2.69 | 71.61±1.34 | 72.90±1.46 | 72.79±1.26 | 74.90±1.33 | 72.58±1.27 | 74.34±1.52 | 75.63±1.59 | 75.40±1.32 | 72.42±1.34 (↑ 3.00) | 76.92±1.26 (↑ 1.29) | 76.57±1.14 (↑ 1.17) |
|  | 20 | 67.61±1.02 | 68.84±2.33 | 67.87±2.22 | 70.00±1.83 | 70.55±2.17 | 71.85±2.22 | 71.99±1.54 | 72.06±1.80 | 72.15±1.27 | 71.32±1.79 | 71.61±0.90 (↑ 4.00) | 74.35±2.17 (↑ 2.20) | 74.44±2.52 (↑ 3.12) |
|  | 25 | 64.81±1.14 | 65.51±2.45 | 66.02±2.39 | 67.89±2.21 | 66.32±2.07 | 68.03±2.39 | 69.47±2.98 | 68.92±2.27 | 69.10±1.33 | 68.72±1.50 | 69.53±2.03 (↑ 4.72) | 71.67±2.07 (↑ 2.57) | 72.07±1.21 (↑ 3.35) |
| Citeseer | 0 | 71.96±0.55 | 73.26±0.83 | 71.20±1.43 | 70.65±0.32 | 72.26±0.83 | 72.10±0.63 | 72.71±0.53 | 72.54±0.21 | 73.26±0.38 | 73.28±0.69 | 73.66±0.83 (↑ 1.70) | 74.21±0.63 (↑ 0.95) | 74.47±0.83 (↑ 1.19) |
|  | 5 | 67.96±0.59 | 68.15±0.89 | 68.20±0.83 | 68.65±1.32 | 69.76±1.48 | 71.32±0.83 | 68.16±1.68 | 69.57±1.38 | 69.36±2.20 | 71.24±1.13 | 69.76±1.38 (↑ 1.80) | 72.10±1.48 (↑ 2.74) | 72.77±0.73 (↑ 1.53) |
|  | 10 | 64.10±1.34 | 66.67±1.35 | 65.92±1.86 | 66.57±1.35 | 66.59±1.98 | 67.86±1.86 | 66.26±1.54 | 68.06±1.46 | 67.52±2.20 | 68.19±1.13 | 67.30±2.89 (↑ 3.20) | 68.53±1.98 (↑ 1.01) | 69.52±2.04 (↑ 1.13) |
|  | 15 | 61.12±1.37 | 63.62±1.47 | 64.64±1.72 | 65.88±1.16 | 65.37±1.34 | 66.43±1.72 | 65.49±2.08 | 66.99±1.25 | 66.89±2.03 | 66.95±1.85 | 65.29±2.99 (↑ 4.17) | 68.25±1.34 (↑ 1.36) | 68.89±1.19 (↑ 1.94) |
|  | 20 | 60.26±1.44 | 61.85±1.35 | 62.93±2.01 | 64.66±1.35 | 63.99±1.44 | 64.00±2.01 | 65.58±1.55 | 65.35±1.44 | 66.37±2.04 | 65.89±2.25 | 63.55±2.64 (↑ 3.29) | 68.17±1.44 (↑ 1.80) | 68.44±1.75 (↑ 2.55) |
|  | 25 | 59.02±1.39 | 59.10±2.24 | 60.87±2.37 | 63.22±2.24 | 63.01±2.26 | 62.85±2.37 | 64.33±2.23 | 64.25±2.23 | 65.11±2.79 | 64.93±1.70 | 63.02±6.28 (↑ 4.00) | 65.24±2.26 (↑ 0.13) | 66.59±2.01 (↑ 1.66) |
| Polblogs | 0 | 95.69±0.38 | 95.35±0.20 | 95.22±0.14 | 95.31±0.18 | 85.13±2.22 | 90.13±2.22 | 95.36±0.82 | 95.13±0.85 | 94.77±1.07 | 93.20±0.64 | 96.13±0.14 (↑ 0.44) | 94.95±2.22 (↑ 0.18) | 95.70±2.22 (↑ 2.50) |
|  | 5 | 87.69±0.38 | 88.35±0.20 | 89.22±0.14 | 90.15±0.18 | 85.03±2.22 | 90.31±0.14 | 89.35±0.32 | 90.05±0.20 | 90.31±1.31 | 90.85±1.39 | 89.72±0.25 (↑ 2.03) | 92.27±2.22 (↑ 1.96) | 92.36±0.07 (↑ 1.51) |
|  | 10 | 84.64±1.69 | 85.88±1.40 | 85.85±1.91 | 86.02±1.33 | 84.56±1.66 | 87.00±1.91 | 87.33±1.74 | 86.68±1.36 | 86.71±1.27 | 87.09±1.41 | 87.71±0.665 (↑ 3.07) | 88.75±1.66 (↑ 2.04) | 89.33±0.23 (↑ 2.24) |
|  | 15 | 71.55±1.70 | 72.01±0.20 | 72.10±1.80 | 78.18±1.89 | 79.89±1.75 | 81.95±1.80 | 82.04±1.47 | 82.12±1.99 | 82.26±1.55 | 83.31±1.87 | 76.58±0.56 (↑ 5.03) | 84.11±1.75 (↑ 1.85) | 84.27±0.27 (↑ 0.96) |
|  | 20 | 65.00±1.03 | 67.12±1.21 | 67.32±1.15 | 69.35±1.32 | 71.25±1.82 | 71.51±1.15 | 69.98±1.22 | 69.42±1.39 | 72.59±2.12 | 72.45±1.69 | 69.44±0.66 (↑ 1.44) | 73.26±1.82 (↑ 1.67) | 73.81±0.26 (↑ 1.36) |
|  | 25 | 64.02±2.35 | 64.16±2.13 | 66.10±2.11 | 67.20±1.44 | 68.39±2.01 | 69.02±2.11 | 66.55±2.79 | 68.22±1.47 | 70.80±2.33 | 69.37±2.01 | 67.92±1.43 (↑ 3.90) | 71.92±2.01 (↑ 1.12) | 72.21±0.43 (↑ 2.84) |
| Pubmed | 0 | 87.19±0.09 | 83.73±0.40 | 86.16±0.18 | 83.44±0.21 | 84.02±0.26 | 87.06±0.06 | 86.74±0.15 | 85.08±0.20 | 87.33±0.18 |  | 88.10±0.18 (↑ 0.91) | 87.15±0.11 (↑ 2.07) | 88.41±0.26 (↑ 1.18) |
|  | 5 | 79.19±0.09 | 80.73±0.40 | 80.16±0.18 | 81.44±0.21 | 80.02±0.26 | 82.44±0.18 | 82.96±0.86 | 82.42±0.21 | 82.17±2.02 | 83.13±1.67 | 81.32±2.08 (↑ 2.13) | 83.15±0.26 (↑ 0.98) | 84.33±1.71 (↑ 1.20) |
|  | 10 | 75.42±1.40 | 76.85±1.59 | 77.93±1.40 | 80.25±1.34 | 78.40±1.70 | 80.45±1.40 | 80.74±1.66 | 80.81±1.30 | 80.59±1.84 | 81.26±1.59 | 78.22±1.43 (↑ 2.80) | 82.10±1.70 (↑ 1.51) | 82.74±1.14 (↑ 1.48) |
|  | 15 | 71.96±1.55 | 72.34±1.55 | 75.75±1.60 | 75.21±1.22 | 73.97±1.76 | 76.89±1.60 | 76.87±1.82 | 76.90±1.25 | 77.31±1.35 | 77.09±2.32 | 75.47±1.34 (↑ 3.51) | 78.49±1.76 (↑ 1.18) | 78.88±1.60 (↑ 1.79) |
|  | 20 | 68.46±0.65 | 68.99±1.72 | 70.03±1.49 | 72.03±1.52 | 71.32±2.60 | 71.62±1.49 | 71.28±1.58 | 73.02±1.52 | 73.20±2.75 | 72.85±2.14 | 73.91±0.90 (↑ 5.45) | 74.81±2.60 (↑ 1.61) | 74.90±2.32 (↑ 2.05) |
|  | 25 | 69.01±2.03 | 68.16±1.01 | 69.02±2.18 | 71.02±2.32 | 70.01±2.51 | 70.86±2.18 | 70.76±2.85 | 72.85±2.31 | 72.69±2.12 | 71.66±2.67 | 72.43±2.03 (↑ 3.42) | 73.96±2.51 (↑ 1.27) | 74.24±1.39 (↑ 2.58) |

the robust GNNs trained on the augmented graph achieve significantly better performance for semi-supervised node classification under graph adversarial attacks. In this section, we directly evaluate the data quality of the synthetic graph structures generated by the RGDM. In the visual domain, the Frechet Inception Distance (FID) is a widely used metric to evaluate the quality of the synthetic images generated by the generative models (Brock et al., 2019; Ho et al., 2020). The FID score measures the similarity between the distribution of the generated images and the distribution of the real images. To compute the FID score, the pre-trained Inception v3 (Szegedy et al., 2016) is used to extract the features from both the real images and the generated images, which are then modeled as the multivariate Gaussian distributions. The FID score is then calculated using the Frechet Distance (FD) (Brock et al., 2019) between the two multivariate Gaussian distributions modeling the real and the generated images (Dowson & Landau, 1982). A lower FID score indicates that the generated images are more similar to the real images, suggesting better quality.

**Quality Evaluation of the Synthetic Nodes.** Although the Inception model cannot be applied to the graph data, we can replace the Inception model in the computation of the FID score with the pre-trained GCN (Kipf & Welling, 2017) for extracting node features to adapt the metric to evaluate the quality of synthetic nodes generated by the RGDM. To this end, we define the Frechet Node Distance (FND), which is the FD between the multivariate Gaussians modeling the node features extracted by pre-trained GCN. We randomly split the nodes in the original clean graph into two partitions of equal size, which are the base partition and the test partition. To mitigate the influence of the randomness, we compute the FND scores with 10 different random splits and report the mean and the standard deviation of the FND scores across different runs. The FND computed between the nodes in the test partition and the base partition of the original clean graph establishes the baseline of the expected FND score for high-quality nodes. By computing the FND score between the features of nodes in the synthetic graph structures generated by the RGDM and the features of nodes in the base partition of the original clean graph, we evaluate the quality of the synthetic nodes in the synthetic graph structures. For simplicity, we refer to the FND score for the synthetic nodes as the

FND between their features and the features of the nodes in the base partition of the original clean graph.

To show the effectiveness of the robustness loss in the training of the RGAE for the RGDM, we also compute the FND for the nodes in the synthetic graph structures generated by the RGDM without the node robustness loss and the edge robustness loss. To demonstrate the advantages of the RGDM over the vanilla diffusion model, the DDPM (Ho et al., 2020), we train a baseline DDPM model on the input node attributes of the original attacked graph and synthesize the same number of synthetic nodes as the RGDM. The synthetic edges are then generated by connecting each synthetic node to its K-nearest neighbors in the original attacked graph using the K-nearest neighbors (KNN) algorithm with $K = \lceil d_{ave} \rceil$, where $d_{ave}$ is the average degree of the original clean graph. The synthetic graph structures, including the synthetic nodes and edges generated by the baseline DDPM model, are combined with the original attacked graph to form the augmented graph. Next, we compute the FND score for the nodes in the synthetic graph structures generated by the baseline DDPM model. In addition, we compute FND for nodes generated by the three baseline node augmentation methods, which are the iGraphMix (Jeong et al., 2024), ImGAGN (Qu et al., 2021), Graphsmote (Zhao et al., 2021a), and SNS (Gao et al., 2023b). We also calculate the FND score for nodes in the original attacked graph. We use the same GCN (Kipf & Welling, 2017) pre-trained on the original clean graph to extract the node features for computing the FND score. The ablation study is performed on the original attacked graph perturbed by the Metattack with a perturbation rate of $25\%$.

Table 8: Frechet Node Distance (FND) to the nodes in the base partition of the original clean graph. The mean and standard deviation of the FND scores computed with 10 different random splits of the base partition and the test partition in the original clean graph are reported. The evaluation is performed for Metattack with a perturbation rate of $25\%$. The FND score for the original clean graph is computed between the nodes in the test partition and the nodes in the base partition of the original clean graph.

| Data | Cora | Citeseer | Polblogs | Pubmed |
|---|---|---|---|---|
| Original Attacked Graph | 12.10±0.45 | 7.53±0.22 | 8.26±0.37 | 6.11±0.38 |
| iGraphMix (Jeong et al., 2024) | 13.44±0.33 | 9.02±0.42 | 10.75±0.44 | 7.65±0.38 |
| ImGAGN (Qu et al., 2021) | 13.10±0.38 | 7.96±0.42 | 9.86±0.43 | 6.95±0.27 |
| Graphsmote (Zhao et al., 2021a) | 13.52±0.42 | 8.95±0.22 | 10.60±0.27 | 7.45±0.51 |
| SNS (Gao et al., 2023b) | 13.35±0.42 | 7.88±0.27 | 9.30±0.85 | 6.74±0.41 |
| Baseline DDPM | 13.28±0.40 | 8.82±0.26 | 10.25±0.39 | 7.25±0.41 |
| RGDM (w/o Robustness Loss) | 11.17±0.46 | 6.15±0.31 | 8.06±0.44 | 5.89±0.35 |
| RGDM (w/o Node Robustness Loss) | 10.02±0.38 | 5.89±0.28 | 6.95±0.35 | 5.01±0.36 |
| RGDM (w/o Edge Robustness Loss) | 9.67±0.27 | 5.18±0.49 | 6.65±0.25 | 5.11±0.38 |
| RGDM | **8.16±0.29** | **4.44±0.32** | **4.35±0.18** | **3.95±0.43** |
| Original Clean Graph | 7.90±0.31 | 4.28±0.25 | 4.16±0.27 | 3.79±0.35 |

The evaluation results are shown in Table 8, where the lower FND scores indicate the node features are more similar to the features of nodes in the base partition of the original clean graph. The FND score of the nodes in the synthetic graph structures generated by the RGDM is closest to the FND score of the original clean graph, which demonstrates that the RGDM generates faithful synthetic nodes. For example, the FND score for the RGDM on Pubmed is 3.95, which is only $4.0\%$ higher than 3.79, which is the FND score of the original clean graph. In addition, we have several key observations from Table 8. First, the synthetic nodes generated by the baseline DDPM are even worse than the original attacked graph. This is because the DDPM is trained on the original attacked graph without any robustness adaptations, and the noises in the original attacked graph are propagated to the synthetic nodes in the augmented graph. Second, the robustness loss is critical in generating clean augmented graphs. Without either the node robustness loss or the edge robustness loss, the synthetic nodes generated by the ablation models are worse than the synthetic nodes generated by the RGDM. This observation proves that both the node robustness loss and the edge robustness loss are beneficial for improving the quality of the synthetic nodes in the synthetic graph structures generated by the RGDM, as mentioned in Section 3.2. Third, the synthetic nodes generated by RGDM are significantly more faithful than those generated by the baseline node-level graph augmentation methods.

**Quality Evaluation of the Synthetic Edges.** Similar to the design of the FND score for evaluating the quality of synthetic nodes, we replace the Inception model in the computation of FID with the

pre-trained GNN (Zhu et al., 2021) for edge feature extraction to adapt the metric to evaluate the quality of the synthetic edges generated by the RGDM. To this end, we define Frechet Edge Distance (FED), which is the FD between the multivariate Gaussians modeling the edge features extracted by pre-trained GNN. Similar to the evaluation of the FND, we randomly split the edges in the original clean graph into two partitions of equal size, which are the base partition and the test partition. To mitigate the influence of the randomness, we compute the FED scores with 10 different random splits and report the mean and the standard deviation of the FED across different runs. The FED computed between the edges in the test partition and the edges in the base partition of the original clean graph establishes the baseline of the expected FED score for the high-quality edges. By computing the FED between the features of edges in the synthetic graph structures generated by the RGDM and the features of edges in the base partition of the original clean graph, we evaluate the quality of the edges in the synthetic graph structures. For simplicity, we refer to the FED score for the synthetic edges as the FED between their features and the features of edges in the base partition of the original clean graph. Similar to the evaluation of the synthetic nodes, we also compute the FED score for the edges in the synthetic graph structures generated by the RGDM without the node robustness loss or the edge robustness loss. We compute the FED score for the edges in the synthetic graph structures generated by the baseline DDPM model. Since the edges in the synthetic graph structures are generated by the KNN algorithm, we evaluate the baseline DDPM models using different values of $K$ from $\{\lceil d_{ave}/4\rceil, \lceil d_{ave}/2\rceil, \lceil d_{ave}\rceil, 2\times\lceil d_{ave}\rceil, 4\times\lceil d_{ave}/4\rceil\}$. In addition, we compute FED for edges generated by the three baseline node augmentation methods, which are the iGraphMix (Jeong et al., 2024), ImGAGN (Qu et al., 2021), Graphsmote (Zhao et al., 2021a), and SNS (Gao et al., 2023b). We also calculate the FED score for nodes in the original attacked graph. The FED score for edges in the original attacked graph is also computed. We use the same NBFNet (Zhu et al., 2021) pre-trained on the original clean graph to extract the edge features for computing the FED score. The ablation study is performed on the original attacked graph perturbed by the Metattack with a perturbation rate of 25%.

Table 9: Frechet Edge Distance (FED) to the nodes in the base partition of the original clean graph. The mean and standard deviation of the FED scores computed with 10 different random splits of the base partition and the test partition in the original clean graph are reported. The evaluation is performed for Metattack with a perturbation rate of 25%. The FED score for the original clean graph is computed between the nodes in the test partition and the nodes in the base partition of the original clean graph.

| Data | Cora | Citeseer | Polblogs | Pubmed |
|---|---|---|---|---|
| Original Attacked Graph | 10.32±0.42 | 7.83±0.28 | 8.92±0.39 | 6.53±0.37 |
| iGraphMix (Jeong et al., 2024) | 11.25±0.28 | 8.42±0.43 | 9.63±0.44 | 7.11±0.27 |
| ImGAGN (Qu et al., 2021) | 11.05±0.38 | 8.04±0.26 | 8.21±0.51 | 7.16±0.79 |
| Graphsmote (Zhao et al., 2021a) | 11.13±0.29 | 8.16±0.51 | 9.32±0.41 | 7.18±0.44 |
| SNS (Gao et al., 2023b) | 11.05±0.38 | 8.04±0.26 | 8.21±0.51 | 7.16±0.79 |
| Baseline DDPM ($K = \lceil d_{ave}/4\rceil$) | 12.03±0.45 | 8.72±0.31 | 10.77±0.40 | 7.93±0.42 |
| Baseline DDPM ($K = \lceil d_{ave}/2\rceil$) | 11.27±0.38 | 8.49±0.27 | 9.32±0.35 | 7.28±0.34 |
| Baseline DDPM ($K = \lceil d_{ave}\rceil$) | 10.39±0.40 | 7.96±0.30 | 8.97±0.37 | 6.83±0.33 |
| Baseline DDPM ($K = \lceil 2 \times d_{ave}\rceil$) | 10.47±0.36 | 7.93±0.29 | 9.21±0.36 | 6.91±0.35 |
| Baseline DDPM ($K = \lceil 4 \times d_{ave}\rceil$) | 10.88±0.41 | 8.01±0.32 | 9.38±0.38 | 7.08±0.38 |
| RGDM (w/o Robustness Loss) | 10.07±0.39 | 7.64±0.26 | 8.74±0.33 | 6.27±0.31 |
| RGDM (w/o Node Robustness Loss) | 8.87±0.35 | 6.73±0.25 | 7.93±0.31 | 5.78±0.32 |
| RGDM (w/o Edge Robustness Loss) | 8.98±0.34 | 6.82±0.27 | 7.72±0.29 | 5.91±0.30 |
| RGDM | **8.34±0.29** | **5.28±0.22** | **6.61±0.25** | **5.34±0.28** |
| Original Clean Graph | 8.03±0.28 | 5.10±0.23 | 6.31±0.24 | 5.15±0.25 |

The evaluation results are shown in Table 9, where the lower FED scores indicate the edge features are more similar to the features of edges in the base partition of the original clean graph. The FED score of the edges in the synthetic graph structures generated by the RGDM is closest to the FED score of the original clean graph, which demonstrates that the RGDM generates faithful synthetic edges. For example, the FED score for the RGDM on Pubmed is 5.34, which is only 3.5% higher than 5.15, which is the FED score of the original clean graph. In addition, we have similar observations from Table 9 as those from Table 8. First, the synthetic edges generated by the baseline DDPM are even worse than the original attacked graph. This is because the DDPM is trained on the original attacked graph without any robustness adaptations, and the simple edge generation

method using the KNN on top of the synthetic nodes generated by the DDPM is vulnerable to noises in the original attacked graph. Edges generation with smaller $K$ in the KNN algorithm is more vulnerable to the noises in the original attacked graph. Although increasing the value of $K$ improves the quality of the synthetic edges, the edges generated by the KNN on top of the nodes generated by the baseline DDPM are still worse than the original attacked graphs. Second, the robustness loss is critical in generating clean augmented graphs. Without either the node robustness loss or the edge robustness loss, the synthetic edges generated by the ablation models are worse than the synthetic edges generated by the RGDM. This observation proves that both the node robustness loss and the edge robustness loss are beneficial for improving the quality of synthetic edges in the synthetic graph structures generated by the RGDM, as mentioned in Section 3.2.

As shown in Table 8 and Table 9, the synthetic nodes and edges in the synthetic graph structures generated by the RGDM have similar FND and FED as the nodes and edges in the original clean graph, which demonstrate that the synthetic graph structures generated by the RGDM are faithful. Therefore, incorporating the synthetic graph structures into the original attacked graph dilutes the adversarial noises. For example, the RGDM generates 912 synthetic edges for Cora under Metattack with a perturbation rate of 25%. After incorporating the synthetic graph structures generated by the RGDM into the original attacked graph, the perturbation rate decreases to 21.1%. In addition, the RGDM generates 785 synthetic edges for Citeseer under Metattack with a perturbation rate of 25%. After incorporating the faithful synthetic graph structures generated by the RGDM into the original attacked graph, the perturbation rate decreases to 20.6%. The dilution of the adversarial noises in these examples provides an explanation of how the RGDM improves the robustness of GNNs trained on the augmented graph. Moreover, the synthetic edges generated by RGDM are significantly more faithful than those generated by the baseline node-level graph augmentation methods.

### F.4 Evaluation on Training Time and Synthetic Data Generation Time

In this section, we detail the training and synthetic data generation times for the Robust Graph Diffusion Model (RGDM) across various datasets in Table 10. The training time for the RGAE and the LDM in the RGDM using a single NVIDIA A100 GPU is measured in minutes. In addition, we present the time for generating synthetic graph structures by the RGDM in seconds per synthetic node generated in Table 10 as well.

Table 10: Time for the training of the RGDM and data generation with the RGDM on different datasets.

| Datasets | RGDM Training Time (minutes) | | Generation Time (s/sample) |
|---|---|---|---|
| | RGAE | LDM | |
| Cora | 20 | 39 | 0.066 |
| Citeseer | 23 | 41 | 0.067 |
| Polblogs | 65 | 31 | 0.069 |
| Pubmed | 153 | 154 | 0.073 |

### F.5 Cross-validation on the Number of Synthetic Nodes

In our experiments, we generate $10 \times |\mathcal{V}_L|$ synthetic nodes, where $V_L$ is the training set of the original graph. Then, we search for the optimal size of synthetic nodes $N'$ added in the augmented graph from $1 \times |\mathcal{V}_L|$ to $10 \times |\mathcal{V}_L|$ with a step size of $|\mathcal{V}_L|$. In the cross-validation, we train different GNN classifiers with different numbers of synthetic data and set $N'$ to the one that achieves the best validation accuracy by the end of 40% of their total epoch number. To study the effectiveness of the cross-validation process for selecting the optimal number of synthetic nodes, we calculate the complete cross-validation time of different models on different datasets. All the experiments are performed on one Nvidia A100 GPU. It is observed from the results in Table 11 that the cross-validation process does not largely increase the computation overhead. The numbers of synthetic nodes added to the augmented graphs for all the experiments are shown in Table 12, Table 13, and Table 14.

Table 11: Time for selecting optimal number synthetic nodes $N'$ with cross-validation.

| Datasets | Cross-validation Time (minutes) | | |
|---|---|---|---|
| | RGDM (GCN) | RGDM (SG-GSR) | RGDM (Pro-GNN) |
| Cora | 1.1 | 3.5 | 43.3 |
| Citeseer | 1.0 | 3.4 | 38.5 |
| Polblogs | 1.6 | 3.9 | 26.9 |
| Pubmed | 9.4 | 20.5 | 176.5 |

Table 12: The number of synthetic nodes $N'$ ($\times |\mathcal{V}_L|$) selected for Metattack, where $V_L$ is the labeled nodes of the original graph.

| Dataset | Ptb Rate (%) | RGDM (GCN) | RGDM (SG-GSR) | RGDM (Pro-GNN) |
|---|---|---|---|---|
| Cora | 0 | $2 \times |\mathcal{V}_L|$ | $1 \times |\mathcal{V}_L|$ | $2 \times |\mathcal{V}_L|$ |
| | 5 | $2 \times |\mathcal{V}_L|$ | $1 \times |\mathcal{V}_L|$ | $2 \times |\mathcal{V}_L|$ |
| | 10 | $2 \times |\mathcal{V}_L|$ | $1 \times |\mathcal{V}_L|$ | $2 \times |\mathcal{V}_L|$ |
| | 15 | $3 \times |\mathcal{V}_L|$ | $1 \times |\mathcal{V}_L|$ | $3 \times |\mathcal{V}_L|$ |
| | 20 | $3 \times |\mathcal{V}_L|$ | $2 \times |\mathcal{V}_L|$ | $3 \times |\mathcal{V}_L|$ |
| | 25 | $5 \times |\mathcal{V}_L|$ | $5 \times |\mathcal{V}_L|$ | $5 \times |\mathcal{V}_L|$ |
| Citeseer | 0 | $1 \times |\mathcal{V}_L|$ | $1 \times |\mathcal{V}_L|$ | $1 \times |\mathcal{V}_L|$ |
| | 5 | $2 \times |\mathcal{V}_L|$ | $2 \times |\mathcal{V}_L|$ | $2 \times |\mathcal{V}_L|$ |
| | 10 | $1 \times |\mathcal{V}_L|$ | $1 \times |\mathcal{V}_L|$ | $2 \times |\mathcal{V}_L|$ |
| | 15 | $3 \times |\mathcal{V}_L|$ | $2 \times |\mathcal{V}_L|$ | $3 \times |\mathcal{V}_L|$ |
| | 20 | $3 \times |\mathcal{V}_L|$ | $3 \times |\mathcal{V}_L|$ | $3 \times |\mathcal{V}_L|$ |
| | 25 | $4 \times |\mathcal{V}_L|$ | $5 \times |\mathcal{V}_L|$ | $4 \times |\mathcal{V}_L|$ |
| Polblogs | 0 | $1 \times |\mathcal{V}_L|$ | $1 \times |\mathcal{V}_L|$ | $1 \times |\mathcal{V}_L|$ |
| | 5 | $1 \times |\mathcal{V}_L|$ | $1 \times |\mathcal{V}_L|$ | $1 \times |\mathcal{V}_L|$ |
| | 10 | $1 \times |\mathcal{V}_L|$ | $2 \times |\mathcal{V}_L|$ | $1 \times |\mathcal{V}_L|$ |
| | 15 | $2 \times |\mathcal{V}_L|$ | $1 \times |\mathcal{V}_L|$ | $2 \times |\mathcal{V}_L|$ |
| | 20 | $2 \times |\mathcal{V}_L|$ | $1 \times |\mathcal{V}_L|$ | $2 \times |\mathcal{V}_L|$ |
| | 25 | $2 \times |\mathcal{V}_L|$ | $2 \times |\mathcal{V}_L|$ | $3 \times |\mathcal{V}_L|$ |
| Pubmed | 0 | $2 \times |\mathcal{V}_L|$ | $1 \times |\mathcal{V}_L|$ | $2 \times |\mathcal{V}_L|$ |
| | 5 | $2 \times |\mathcal{V}_L|$ | $1 \times |\mathcal{V}_L|$ | $2 \times |\mathcal{V}_L|$ |
| | 10 | $3 \times |\mathcal{V}_L|$ | $1 \times |\mathcal{V}_L|$ | $2 \times |\mathcal{V}_L|$ |
| | 15 | $3 \times |\mathcal{V}_L|$ | $1 \times |\mathcal{V}_L|$ | $2 \times |\mathcal{V}_L|$ |
| | 20 | $3 \times |\mathcal{V}_L|$ | $1 \times |\mathcal{V}_L|$ | $3 \times |\mathcal{V}_L|$ |
| | 25 | $2 \times |\mathcal{V}_L|$ | $1 \times |\mathcal{V}_L|$ | $3 \times |\mathcal{V}_L|$ |

Table 13: The number of synthetic nodes $N'$ ($\times |\mathcal{V}_L|$) for Nettack, where $V_L$ is the labeled nodes of the original graph.

| Dataset | Attack Budget | RGDM (GCN) | RGDM (SG-GSR) | RGDM (Pro-GNN) |
|---------|---------------|------------|---------------|----------------|
| Cora | 0 | $2 \times \vert\mathcal{V}_L\vert$ | $1 \times \vert\mathcal{V}_L\vert$ | $2 \times \vert\mathcal{V}_L\vert$ |
| | 1 | $3 \times \vert\mathcal{V}_L\vert$ | $2 \times \vert\mathcal{V}_L\vert$ | $2 \times \vert\mathcal{V}_L\vert$ |
| | 2 | $3 \times \vert\mathcal{V}_L\vert$ | $2 \times \vert\mathcal{V}_L\vert$ | $2 \times \vert\mathcal{V}_L\vert$ |
| | 3 | $3 \times \vert\mathcal{V}_L\vert$ | $2 \times \vert\mathcal{V}_L\vert$ | $3 \times \vert\mathcal{V}_L\vert$ |
| | 4 | $3 \times \vert\mathcal{V}_L\vert$ | $3 \times \vert\mathcal{V}_L\vert$ | $3 \times \vert\mathcal{V}_L\vert$ |
| | 5 | $4 \times \vert\mathcal{V}_L\vert$ | $3 \times \vert\mathcal{V}_L\vert$ | $3 \times \vert\mathcal{V}_L\vert$ |
| Citeseer | 0 | $2 \times \vert\mathcal{V}_L\vert$ | $1 \times \vert\mathcal{V}_L\vert$ | $1 \times \vert\mathcal{V}_L\vert$ |
| | 1 | $2 \times \vert\mathcal{V}_L\vert$ | $2 \times \vert\mathcal{V}_L\vert$ | $2 \times \vert\mathcal{V}_L\vert$ |
| | 2 | $1 \times \vert\mathcal{V}_L\vert$ | $2 \times \vert\mathcal{V}_L\vert$ | $2 \times \vert\mathcal{V}_L\vert$ |
| | 3 | $3 \times \vert\mathcal{V}_L\vert$ | $3 \times \vert\mathcal{V}_L\vert$ | $3 \times \vert\mathcal{V}_L\vert$ |
| | 4 | $3 \times \vert\mathcal{V}_L\vert$ | $2 \times \vert\mathcal{V}_L\vert$ | $3 \times \vert\mathcal{V}_L\vert$ |
| | 5 | $3 \times \vert\mathcal{V}_L\vert$ | $3 \times \vert\mathcal{V}_L\vert$ | $3 \times \vert\mathcal{V}_L\vert$ |
| Polblogs | 0 | $1 \times \vert\mathcal{V}_L\vert$ | $1 \times \vert\mathcal{V}_L\vert$ | $1 \times \vert\mathcal{V}_L\vert$ |
| | 1 | $1 \times \vert\mathcal{V}_L\vert$ | $1 \times \vert\mathcal{V}_L\vert$ | $1 \times \vert\mathcal{V}_L\vert$ |
| | 2 | $1 \times \vert\mathcal{V}_L\vert$ | $2 \times \vert\mathcal{V}_L\vert$ | $1 \times \vert\mathcal{V}_L\vert$ |
| | 3 | $1 \times \vert\mathcal{V}_L\vert$ | $2 \times \vert\mathcal{V}_L\vert$ | $2 \times \vert\mathcal{V}_L\vert$ |
| | 4 | $2 \times \vert\mathcal{V}_L\vert$ | $2 \times \vert\mathcal{V}_L\vert$ | $2 \times \vert\mathcal{V}_L\vert$ |
| | 5 | $2 \times \vert\mathcal{V}_L\vert$ | $3 \times \vert\mathcal{V}_L\vert$ | $3 \times \vert\mathcal{V}_L\vert$ |
| Pubmed | 0 | $1 \times \vert\mathcal{V}_L\vert$ | $1 \times \vert\mathcal{V}_L\vert$ | $2 \times \vert\mathcal{V}_L\vert$ |
| | 1 | $1 \times \vert\mathcal{V}_L\vert$ | $1 \times \vert\mathcal{V}_L\vert$ | $2 \times \vert\mathcal{V}_L\vert$ |
| | 2 | $2 \times \vert\mathcal{V}_L\vert$ | $2 \times \vert\mathcal{V}_L\vert$ | $2 \times \vert\mathcal{V}_L\vert$ |
| | 3 | $2 \times \vert\mathcal{V}_L\vert$ | $2 \times \vert\mathcal{V}_L\vert$ | $2 \times \vert\mathcal{V}_L\vert$ |
| | 4 | $2 \times \vert\mathcal{V}_L\vert$ | $3 \times \vert\mathcal{V}_L\vert$ | $3 \times \vert\mathcal{V}_L\vert$ |
| | 5 | $2 \times \vert\mathcal{V}_L\vert$ | $3 \times \vert\mathcal{V}_L\vert$ | $3 \times \vert\mathcal{V}_L\vert$ |

Table 14: The number of synthetic nodes $N'$ ($\times |\mathcal{V}_L|$) for Topology Attack, where $V_L$ is the labeled nodes of the original graph.

| Dataset | Ptb Rate (%) | RGDM (GCN) | RGDM (SG-GSR) | RGDM (Pro-GNN) |
|---------|--------------|------------|---------------|----------------|
| Cora | 0 | $1 \times \vert\mathcal{V}_L\vert$ | $1 \times \vert\mathcal{V}_L\vert$ | $1 \times \vert\mathcal{V}_L\vert$ |
| | 5 | $1 \times \vert\mathcal{V}_L\vert$ | $1 \times \vert\mathcal{V}_L\vert$ | $1 \times \vert\mathcal{V}_L\vert$ |
| | 10 | $2 \times \vert\mathcal{V}_L\vert$ | $2 \times \vert\mathcal{V}_L\vert$ | $2 \times \vert\mathcal{V}_L\vert$ |
| | 15 | $2 \times \vert\mathcal{V}_L\vert$ | $1 \times \vert\mathcal{V}_L\vert$ | $2 \times \vert\mathcal{V}_L\vert$ |
| | 20 | $2 \times \vert\mathcal{V}_L\vert$ | $2 \times \vert\mathcal{V}_L\vert$ | $3 \times \vert\mathcal{V}_L\vert$ |
| | 25 | $2 \times \vert\mathcal{V}_L\vert$ | $2 \times \vert\mathcal{V}_L\vert$ | $2 \times \vert\mathcal{V}_L\vert$ |
| Citeseer | 0 | $1 \times \vert\mathcal{V}_L\vert$ | $1 \times \vert\mathcal{V}_L\vert$ | $1 \times \vert\mathcal{V}_L\vert$ |
| | 5 | $2 \times \vert\mathcal{V}_L\vert$ | $1 \times \vert\mathcal{V}_L\vert$ | $1 \times \vert\mathcal{V}_L\vert$ |
| | 10 | $1 \times \vert\mathcal{V}_L\vert$ | $2 \times \vert\mathcal{V}_L\vert$ | $2 \times \vert\mathcal{V}_L\vert$ |
| | 15 | $2 \times \vert\mathcal{V}_L\vert$ | $2 \times \vert\mathcal{V}_L\vert$ | $3 \times \vert\mathcal{V}_L\vert$ |
| | 20 | $2 \times \vert\mathcal{V}_L\vert$ | $2 \times \vert\mathcal{V}_L\vert$ | $2 \times \vert\mathcal{V}_L\vert$ |
| | 25 | $3 \times \vert\mathcal{V}_L\vert$ | $2 \times \vert\mathcal{V}_L\vert$ | $2 \times \vert\mathcal{V}_L\vert$ |
| Polblogs | 0 | $1 \times \vert\mathcal{V}_L\vert$ | $1 \times \vert\mathcal{V}_L\vert$ | $1 \times \vert\mathcal{V}_L\vert$ |
| | 5 | $1 \times \vert\mathcal{V}_L\vert$ | $1 \times \vert\mathcal{V}_L\vert$ | $1 \times \vert\mathcal{V}_L\vert$ |
| | 10 | $1 \times \vert\mathcal{V}_L\vert$ | $2 \times \vert\mathcal{V}_L\vert$ | $2 \times \vert\mathcal{V}_L\vert$ |
| | 15 | $2 \times \vert\mathcal{V}_L\vert$ | $2 \times \vert\mathcal{V}_L\vert$ | $3 \times \vert\mathcal{V}_L\vert$ |
| | 20 | $2 \times \vert\mathcal{V}_L\vert$ | $3 \times \vert\mathcal{V}_L\vert$ | $1 \times \vert\mathcal{V}_L\vert$ |
| | 25 | $2 \times \vert\mathcal{V}_L\vert$ | $2 \times \vert\mathcal{V}_L\vert$ | $2 \times \vert\mathcal{V}_L\vert$ |
| Pubmed | 0 | $1 \times \vert\mathcal{V}_L\vert$ | $1 \times \vert\mathcal{V}_L\vert$ | $2 \times \vert\mathcal{V}_L\vert$ |
| | 5 | $2 \times \vert\mathcal{V}_L\vert$ | $2 \times \vert\mathcal{V}_L\vert$ | $3 \times \vert\mathcal{V}_L\vert$ |
| | 10 | $2 \times \vert\mathcal{V}_L\vert$ | $2 \times \vert\mathcal{V}_L\vert$ | $2 \times \vert\mathcal{V}_L\vert$ |
| | 15 | $3 \times \vert\mathcal{V}_L\vert$ | $1 \times \vert\mathcal{V}_L\vert$ | $2 \times \vert\mathcal{V}_L\vert$ |
| | 20 | $2 \times \vert\mathcal{V}_L\vert$ | $3 \times \vert\mathcal{V}_L\vert$ | $4 \times \vert\mathcal{V}_L\vert$ |
| | 25 | $1 \times \vert\mathcal{V}_L\vert$ | $2 \times \vert\mathcal{V}_L\vert$ | $2 \times \vert\mathcal{V}_L\vert$ |

## F.6 ADVERSARIAL DEFENSE ON LARGE SCALE GRAPHS

To evaluate the scalability of the proposed method, we conduct additional experiments on graph adversarial defense for large-scale graph datasets, which are the large-scale social network, Reddit (Hamilton et al., 2017), and the large-scale knowledge graph, ogbn-arxiv (Hu et al., 2020). The statistics of the Reddit and ogbn-arxiv datasets are detailed in Table 4. Since the Metattack (Zügner & Günnemann, 2019), the Nettack (Zügner et al., 2018), and the Topology Attack (Xu et al., 2019) adopted in Section 4.2 do not scale to large graphs at the size of Reddit and ogbn-arxiv, we use a scalable graph adversarial attack method, the improved DICE (Li et al., 2022c), in our experi-

ments on Reddit and ogbn-arxiv. The experiments are performed for perturbation rates of $0\%$, $10\%$, and $20\%$ following the settings in (Li et al., 2022c) with GCN as the baseline model. The results are shown in Table 15. It is observed that RGDM consistently improves the performance of the GCN baseline across all perturbation levels on both datasets. For example, RGDM achieves an improvement of $2.25\%$ on ogbn-arxiv at the perturbation rate of $10\%$, indicating its effectiveness in defending against adversarial perturbations even on large and complex knowledge graphs.

Table 15: Node classification performance (Accuracy±Std) under the adversarial attack by the improved DICE (Li et al., 2022c). The improvements of the RGDM over the baseline are attached in parentheses.

| Dataset | Ptb Rate (%) | GCN | RGDM (GCN) |
|---------|:---:|:---:|:---:|
| ogbn-arxiv | 0 | 72.82±1.32 | **74.48** ±1.15 (↑ 1.66) |
| | 10 | 62.25±1.23 | **64.50** ±1.35 (↑ 2.25) |
| | 20 | 54.95±1.44 | **56.17** ±1.66 (↑ 1.22) |
| Reddit | 0 | 93.52±1.17 | **95.44** ±1.42 (↑ 1.92) |
| | 10 | 82.62±1.32 | **84.73** ±1.12 (↑ 2.11) |
| | 20 | 74.53±1.57 | **75.99** ±1.22 (↑ 1.46) |

### F.7 SENSITIVITY ANALYSIS ON THE HYPER-PARAMETERS

In this section, we conduct a sensitivity analysis on the temperature parameter $\tau$ for edge selection. The study is carried out using RGDM (Pro-GNN) on the Cora dataset under a $25\%$ Metattack setting. As shown in Table 16, although the highest accuracy is achieved at $\tau = 0.5$, the performance remains stable across different values of $\tau$. In particular, even the lowest performing setting, $\tau = 0.1$, results in only a marginal $0.19\%$ decrease in accuracy compared to the best result.

Table 16: Sensitivity analysis on the temperature for edge selection $\tau$. The study is performed by using RGDM (Pro-GNN) on Cora with $25\%$ of Metattack.

| $\tau$ | 0.1 | 0.2 | 0.3 | 0.4 | 0.5 | 0.6 | 0.7 | 0.8 | 0.9 |
|---------|---|---|---|---|---|---|---|---|---|
| Accuracy | 72.12 | 72.16 | 72.30 | 72.29 | 72.31 | 72.28 | 72.24 | 72.22 | 72.25 |

### F.8 ADVERSARIAL DEFENSE ON HETEROPHILIC GRAPH

In this section, we study the effectiveness of RGDM for adversarial defense on heterophilic graphs. Following (Qiu et al., 2024), we perform the study on Chameleon (Pei et al., 2020) and Actor (Tang et al., 2009) for the defense against the Metattack (Zügner & Günnemann, 2019). The adversarial robust GNN, LHS (Qiu et al., 2024), which is specifically designed for heterophilic graphs, is adopted as the baseline model. The results are shown in Table 17. It is observed that RGDM significantly improves the performance of LHS. For example, RGDM achieves an improvement of $2.38\%$ over the baseline model on Chameleon under a perturbation rate of $25\%$.

Table 17: Node classification performance (Accuracy±Std) under the non-targeted poisoning attack (Metattack) (Zügner & Günnemann, 2019). The improvements of the RGDM over the baseline are attached in parentheses.

| Dataset | Ptb Rate (%) | LHS | RGDM (LHS) |
|---------|:---:|:---:|:---:|
| Chameleon | 0 | 72.31±1.32 | **74.02** ±2.13 (↑ 1.71) |
| | 15 | 71.87±1.61 | **72.98** ±2.11 (↑ 1.11) |
| | 25 | 70.03±1.85 | **72.41** ±1.66 (↑ 2.38) |
| Actor | 0 | 38.87±1.43 | **41.80** ±1.42 (↑ 2.93) |
| | 15 | 36.79±1.11 | **39.01** ±1.12 (↑ 2.22) |
| | 25 | 35.37±2.12 | **37.90** ±2.01 (↑ 2.53) |

## F.9    t-SNE Visualization of the Augmented Graph

Figure 5 illustrates the t-SNE visualization of both the real and synthetic node features within the augmented graph, revealing that the synthetic nodes closely mimic the real nodes.

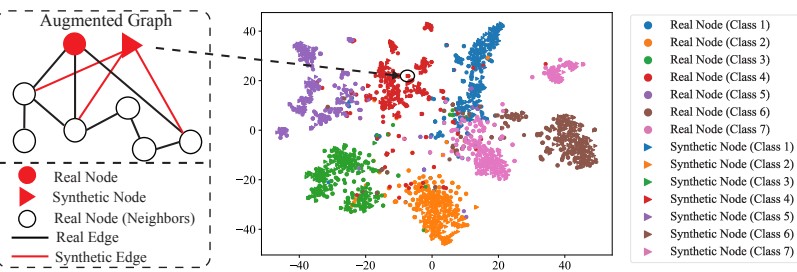

Figure 5: Illustration of the augmented graph after adding the synthetic graph structures into the original graph on Cora. The RGDM is trained on Cora under Mettack with a perturbation rate of 25%. The figure on the right illustrates the t-SNE visualization of the node features for both the real nodes in the original clean graph and the synthetic nodes. The figure on the left illustrates the 2-hop neighborhood of a real node and a synthetic node that have similar node features, evidenced by the t-SNE visualization in the augmented graph. It is observed that the synthetic node shares the same neighbors as the real node.

## F.10    Comparison with Existing Node-Level Graph Augmentation Methods

To demonstrate the superiority of RGDM over existing graph augmentation methods, we compare RGDM with mix-up based conventional graph augmentation method, iGraphMix (Jeong et al., 2024), and GAN-based SGS generation methods, ImGAGN (Qu et al., 2021), Graphsmote (Zhao et al., 2021a), and SNS (Gao et al., 2023b). The comparison is performed on Cora, Citeseer, Polblogs, and Pubmed under the Metattack with the perturbation rate of 25%. For a fair comparison, we compare the performance of the GCN augmented by the competing node-level graph augmentation methods, iGraphMix, ImGAGN, Graphsmote, and SNS, with RGDM (GCN). The number of synthetic nodes added by each of the baseline methods is also selected by cross-validation following the settings in Section F.5 It is observed in Table 18 that RGDM (GCN) significantly outperforms the GCNs augmented by all the competing baseline methods. For instance, RGDM (GCN) outperforms SNS (GCN) by 3.24% on the PubMed dataset, demonstrating the advantages of the proposed RGDM in improving the robustness of the GNNs trained on the augmented graph generated by it.

Table 18: Node classification accuracy (%) comparisons between RGDM and existing node-level graph augmentation methods. The study is performed for Metattack with a perturbation rate of 25%.

| Method | Cora | Citeseer | Polblogs | Pubmed |
|---|---|---|---|---|
| GCN | 47.53 | 56.94 | 49.23 | 75.50 |
| iGraphMix (GCN) (Jeong et al., 2024) | 53.25 | 57.11 | 51.65 | 76.10 |
| ImGAGN (GCN) (Qu et al., 2021) | 53.58 | 56.89 | 50.95 | 75.32 |
| Graphsmote (GCN) (Zhao et al., 2021a) | 55.10 | 58.06 | 52.44 | 76.25 |
| SNS (GCN) (Gao et al., 2023b) | 58.35 | 59.17 | 54.03 | 78.28 |
| RGDM (GCN) | **61.42** | **62.84** | **56.47** | **81.52** |

## F.11    Statistical Significance Analysis

In this section, we calculate the p-values of the t-tests for RGDM (GCN), RGDM (Pro-GNN), and RGDM (SG-GSR) against their corresponding methods GCN, Pro-GNN, and SG-GSR to evaluate the statistical significance of the improvements by RGDM. It is observed in Table 19 that the p-values for both GCL-LR and GCL-LRA against the best baseline methods across all datasets and noise settings are consistently smaller than 0.05, indicating statistically significant improvements over the top baseline methods.

Table 19: P-values of the t-tests for RGDM (GCN), RGDM (Pro-GNN), and RGDM (SG-GSR) against their corresponding methods GCN, Pro-GNN, and SG-GSR.

| Dataset | Metattack | | | | Netattack | | | | Topology Attack | | | |
|---|---|---|---|---|---|---|---|---|---|---|---|---|
| | Ptb Rate (%) | RGDM (GCN) | RGDM (Pro-GNN) | RGDM (SG-GSR) | Attack Budget | RGDM (GCN) | RGDM (Pro-GNN) | RGDM (SG-GSR) | Ptb Rate (%) | RGDM (GCN) | RGDM (Pro-GNN) | RGDM (SG-GSR) |
| Cora | 0 | 0.042 | 0.027 | 0.016 | 0 | 0.041 | 0.025 | 0.015 | 0 | 0.043 | 0.029 | 0.017 |
| | 5 | 0.038 | 0.024 | 0.013 | 1 | 0.036 | 0.023 | 0.011 | 5 | 0.039 | 0.025 | 0.012 |
| | 10 | 0.033 | 0.021 | 0.012 | 2 | 0.032 | 0.019 | 0.010 | 10 | 0.035 | 0.020 | 0.011 |
| | 15 | 0.029 | 0.017 | 0.010 | 3 | 0.027 | 0.016 | 0.008 | 15 | 0.030 | 0.018 | 0.009 |
| | 20 | 0.025 | 0.015 | 0.009 | 4 | 0.022 | 0.013 | 0.007 | 20 | 0.026 | 0.014 | 0.008 |
| | 25 | 0.021 | 0.012 | 0.008 | 5 | 0.019 | 0.011 | 0.006 | 25 | 0.022 | 0.012 | 0.007 |
| Citeseer | 0 | 0.039 | 0.022 | 0.018 | 0 | 0.037 | 0.021 | 0.016 | 0 | 0.038 | 0.023 | 0.017 |
| | 5 | 0.035 | 0.020 | 0.015 | 1 | 0.033 | 0.019 | 0.013 | 5 | 0.034 | 0.020 | 0.014 |
| | 10 | 0.030 | 0.017 | 0.013 | 2 | 0.028 | 0.016 | 0.011 | 10 | 0.029 | 0.017 | 0.012 |
| | 15 | 0.027 | 0.014 | 0.011 | 3 | 0.024 | 0.013 | 0.009 | 15 | 0.026 | 0.014 | 0.010 |
| | 20 | 0.023 | 0.012 | 0.009 | 4 | 0.021 | 0.011 | 0.007 | 20 | 0.022 | 0.012 | 0.008 |
| | 25 | 0.019 | 0.010 | 0.007 | 5 | 0.018 | 0.009 | 0.006 | 25 | 0.020 | 0.010 | 0.007 |
| Polblogs | 0 | 0.041 | 0.023 | 0.019 | 0 | 0.040 | 0.021 | 0.017 | 0 | 0.042 | 0.022 | 0.018 |
| | 5 | 0.036 | 0.020 | 0.015 | 1 | 0.034 | 0.018 | 0.013 | 5 | 0.035 | 0.019 | 0.014 |
| | 10 | 0.031 | 0.017 | 0.013 | 2 | 0.029 | 0.016 | 0.011 | 10 | 0.030 | 0.017 | 0.012 |
| | 15 | 0.027 | 0.014 | 0.011 | 3 | 0.025 | 0.013 | 0.009 | 15 | 0.026 | 0.015 | 0.010 |
| | 20 | 0.024 | 0.011 | 0.009 | 4 | 0.022 | 0.010 | 0.007 | 20 | 0.023 | 0.011 | 0.008 |
| | 25 | 0.020 | 0.009 | 0.006 | 5 | 0.019 | 0.008 | 0.005 | 25 | 0.021 | 0.009 | 0.006 |
| Pubmed | 0 | 0.038 | 0.021 | 0.017 | 0 | 0.037 | 0.020 | 0.015 | 0 | 0.039 | 0.022 | 0.016 |
| | 5 | 0.034 | 0.018 | 0.014 | 1 | 0.033 | 0.017 | 0.012 | 5 | 0.034 | 0.018 | 0.013 |
| | 10 | 0.030 | 0.015 | 0.012 | 2 | 0.029 | 0.014 | 0.010 | 10 | 0.030 | 0.015 | 0.011 |
| | 15 | 0.027 | 0.013 | 0.010 | 3 | 0.026 | 0.012 | 0.008 | 15 | 0.027 | 0.013 | 0.009 |
| | 20 | 0.023 | 0.010 | 0.008 | 4 | 0.021 | 0.009 | 0.006 | 20 | 0.022 | 0.010 | 0.007 |
| | 25 | 0.019 | 0.008 | 0.006 | 5 | 0.018 | 0.007 | 0.005 | 25 | 0.019 | 0.008 | 0.006 |

### F.12 STUDY ON RGDM FOR GRAPH PURIFICATION

The RGDM proposed in Section 3 has demonstrated superior performance for generative data augmentation (GDA), which involves generating faithful synthetic graph structures and integrating them into the original attacked graph to obtain an augmented graph. In this section, we study the effectiveness of applying RGDM to purify the attacked graph. The graph purification by RGDM, referred to as RGDM (Purification), is achieved by reconstructing each node and its corresponding edges through the LDM, where Gaussian noise derived from the original node and its edges is progressively denoised by the LDM and subsequently decoded by the GAE to produce clean node features and edges. The RGDM (Purification) variant only purifies the original graph, aiming to explore whether RGDM can be adapted to directly denoise/remove the adversarial perturbations in the original attacked graph. Moreover, the RGDM (Purification) can be combined with RGDM for GDA, denoted as RGDM (GDA) here, leading to a variant that performs both GDA and graph purification, denoted as RGDM (GDA + Purification). The study is performed on Cora under Metattack with a 25% attack rate. Table 20 shows that applying RGDM for purification enhances the performance of Pro-GNN, while GDA with RGDM achieves better performance. Moreover, it is observed that combining GDA and graph purification by RGDM further improves the performance.

Table 20: Study on the effectiveness of graph purification and GDA by RGDM. The evaluation is conducted on Cora under Metattack with 25% perturbation rate.

| Method | Accuracy (%) |
|---|---|
| Pro-GNN | 69.7 |
| RGDM (Purification) | 71.0 |
| RGDM (GDA) | 72.3 |
| RGDM (GDA + Purification) | **72.9** |

### F.13 COMPARISON WITH DIFFUSION-BASED GRAPH PURIFICATION METHODS

To demonstrate the superiority of RGDM over existing diffusion-based graph purification methods (He et al., 2025; Luo et al., 2025), we have performed experiments comparing RGDM with GDDM (He et al., 2025) and DiffSP (Luo et al., 2025) on Cora, Citeseer, Polblogs, and Pubmed under Metattack with a perturbation rate of 25%. We run each experiment 10 times with different random seeds and report the mean and the standard deviation of the node classification accuracy. We use SG-GSR (In et al., 2024), which is the state-of-the-art robust GNN model, as the baseline model for RGDM following the settings in Section 4.2 of the main paper. The GDDM and DiffSP are used to purify the input graph to SG-GSR, and our proposed RGDM is used to augment the input graph to SG-GSR with synthetic graph structures. It is observed in Table 21 that RGDM (SG-GSR) outperforms DiffSP and GDDM on all the datasets. To validate the significance of the improvements, we calculate the p-values of the t-tests between RGDM (SG-GSR) and the best-performing baseline method on each of the datasets. As observed in the table below, the p-values for RGDM (SG-GSR) against the best baseline methods across all datasets are consistently smaller than 0.05, indicating

statistically significant improvements of RGDM (SG-GSR) over the top baseline methods among DiffSP and GDDM.

Table 21: Comparison of classification accuracy ($\pm$Std) and p-values across different datasets between RGDM and diffusion-based graph purification methods.

| Datasets | GDDM (He et al., 2025) | DiffSP (Luo et al., 2025) | RGDM (SG-GSR) | p-value |
|---|---|---|---|---|
| Cora | $77.2 \pm 0.3$ | $76.8 \pm 0.5$ | $\mathbf{79.1 \pm 0.4}$ | 0.0022 |
| Citeseer | $71.6 \pm 0.5$ | $71.8 \pm 0.2$ | $\mathbf{75.3 \pm 0.7}$ | 0.0012 |
| Polblogs | $84.8 \pm 0.8$ | $85.6 \pm 0.5$ | $\mathbf{88.8 \pm 0.6}$ | 0.0004 |
| Pubmed | $85.9 \pm 0.6$ | $86.8 \pm 0.4$ | $\mathbf{88.6 \pm 0.5}$ | 0.0017 |

### F.14   ABLATION ON THE NUMBER OF SYNTHETIC NODES

To assess the effect of the number of synthetic nodes on robustness, we conduct an ablation study by varying the number of synthetic nodes from 0 to $10 \times |\mathcal{V}_L|$. We evaluate the performance of RGDM with Pro-GNN as the base GNN for node classification under Metattack with a perturbation rate of 25%. As shown in Table 22, increasing the number of synthetic nodes initially improves performance, with the best result achieved at $5 \times |\mathcal{V}_L|$. However, further increasing the number of synthetic nodes leads to slightly degraded performance.

Table 22: Performance of RGDM (Pro-GNN) under Metattack with varying numbers of synthetic nodes. The perturbation rate is 25%.

| # Synthetic Nodes ($\times|\mathcal{V}_L|$) | 1 | 2 | 3 | 4 | 5 | 6 | 7 | 8 | 9 | 10 |
|---|---|---|---|---|---|---|---|---|---|---|
| Accuracy (%) | 70.5 | 71.0 | 71.9 | 72.2 | **72.3** | 72.1 | 72.0 | 71.7 | 71.4 | 71.2 |

### F.15   PERFORMANCE COMPARISON WITH VANILLA DDPM-AUGMENTED GRAPHS

To demonstrate the superiority of the synthetic graph structures generated by RGDM, we compare RGDM (Pro-GNN) with two baselines, which are Pro-GNN trained on the original attacked graph and Pro-GNN trained on a graph augmented with synthetic nodes generated by a vanilla DDPM. The vanilla DDPM baseline synthesizes labeled synthetic nodes via conditional DDPM and connects them using $k$-nearest neighbors (KNN), as detailed in Section F.3. The study is performed on Cora under Metattack with a perturbation rate of 25%. As shown in Table 23, the vanilla DDPM even degrades performance, confirming that RGDM's gains stem from the generated faithful synthetic graph structures, rather than simply increasing the number of training nodes/edges.

Table 23: Performance comparison of Pro-GNN on augmented graph with synthetic graph structures generated by DDPM and RGDM. The study is performed on Cora under Metattack with a perturbation rate of 25%.

| Method | Accuracy (%) |
|---|---|
| Pro-GNN | 69.7 |
| DDPM (Pro-GNN) | 68.2 |
| RGDM (Pro-GNN) | **72.3** |

### F.16   STUDY ON THE IMPACT OF LABEL RATIO

To assess the robustness of RGDM (Pro-GNN) under varying levels of supervision, we evaluate performance on the Cora dataset under a 25% Metattack perturbation rate while varying the labeled training ratio. In this study, we consider both sparse label regimes (2.5% to 10%) and denser settings (20% to 60%), with 10% of nodes fixed for validation and the remainder used for testing. As shown in Table 24, RGDM consistently outperforms the Pro-GNN baseline across all label ratios. Under extremely limited supervision (2.5%–10%), RGDM maintains strong performance, indicating its utility as a data augmentation strategy in low-resource scenarios. As the label ratio increases,

RGDM continues to achieve better performance, demonstrating its robustness and effectiveness even with more abundant training data. These results confirm that the improvements of RGDM are not confined to a specific supervision regime and generalize well across both low- and high-label density settings.

Table 24: Performance comparison of Pro-GNN and RGDM (Pro-GNN) under different label ratios on Cora under $25\%$ Metattack.

| Label Ratio | 2.5% | 5% | 7.5% | 10% | 20% | 40% | 60% |
|---|---|---|---|---|---|---|---|
| Pro-GNN | 62.5 | 65.6 | 67.5 | 69.7 | 73.8 | 75.2 | 76.6 |
| RGDM (Pro-GNN) | **64.8** | **68.0** | **69.3** | **72.3** | **75.9** | **77.6** | **78.4** |

### F.17 ABLATION STUDY ON THE SPARSE HIERARCHICAL EDGE DECODER

To further validate the effectiveness of the proposed Sparse Hierarchical Edge Decoder (SHED) in generating robust synthetic edges, we conduct an ablation study comparing SHED against the Bi-Level Neighborhood Decoder (BLND) used in DoG (Wang et al., 2025). In particular, we replace SHED in RGDM with the BLND module and denote the resulting model as RGDM (BLND). Both models are evaluated under the Metattack (Zügner & Günnemann, 2019) with a perturbation rate of $25\%$, using SG-GSR (In et al., 2024) as the backbone GNN. The experiments are conducted on Cora, Citeseer, Polblogs, and Pubmed. As shown in Table 25, RGDM equipped with SHED consistently outperforms RGDM (BLND) across all benchmarks. For instance, RGDM achieves an improvement of $2.0\%$ over RGDM (BLND) on the Pubmed dataset, demonstrating the superiority of SHED to reconstruct clean edge structures in the presence of adversarial noise in the given graph.

Table 25: Performance comparison between the proposed Sparse Hierarchical Edge Decoder (SHED) and the Bi-Level Neighborhood Decoder (BLND) from DoG (Wang et al., 2025). SG-GSR is used as the base GNN under Metattack with $25\%$ perturbation.

| Dataset | Cora | Citeseer | Polblogs | Pubmed |
|---|---|---|---|---|
| RGDM (BLND) | 77.4 | 73.1 | 87.2 | 86.6 |
| RGDM | **79.1** | **75.3** | **88.8** | **88.6** |

