# OpenReview forum: "Robust Graph Diffusion Model"
_ICLR.cc/2026/Conference — ICLR 2026 Conference Withdrawn Submission_

### Official Review · Reviewer_93G4 · 2025-10-25

**Soundness:** 2
**Presentation:** 3
**Contribution:** 2
**Rating:** 4
**Confidence:** 4

**Summary:**

This paper introduces the RGDM, a novel framework designed to enhance the robustness of GNNs against adversarial attacks. The RGDM framework aims to generate high-quality synthetic graph structures that are then merged with the attacked graph to "dilute" adversarial noise. The approach includes a RGAE with low-rank regularization to learn a purified latent representation and a Latent Diffusion Model to generate new data points in this latent space. The paper also proposes a Sparse Hierarchical Edge Decoder for efficient edge generation.

**Strengths:**

1.The use of diffusion models for node-level graph augmentation is a fresh and innovative approach. The "augment-and-dilute" strategy offers a compelling alternative to conventional defense methods like adversarial training or graph purification.

2.The RGDM framework is technically sophisticated and thoughtfully designed.

3.The experimental validation is comprehensive.

**Weaknesses:**

1.The RGDM framework is complex, involving multiple stages and interconnected components. This raises concerns about the practicality of reproducing the results and the ease of adoption in real-world applications.

2.While the paper provides some details on generation time, it would be useful to see a direct comparison of the total training cost. The RGDM pipeline (especially the diffusion model) likely incurs high computational costs, and a clearer discussion of the trade-off between the cost of training and the performance gains would be valuable.

**Questions:**

1.The low-rank regularization is shown to be crucial in the ablation study (Table 3). How sensitive is the model’s performance to the choice of the rank parameter, ro? Could a very low rank risk filtering out high-frequency signals that may be important for the task at hand?

2.The defense strategy relies on "diluting" the attack. How would RGDM perform against an adaptive attack, where the adversary is aware of the defense mechanism and might craft perturbations that target the synthetic nodes or edges generated by the model?

3.Can you provide a more direct comparison of the end-to-end training time?

---

### Official Review · Reviewer_7rCF · 2025-11-01

**Soundness:** 3
**Presentation:** 2
**Contribution:** 2
**Rating:** 2
**Confidence:** 3

**Summary:**

This paper introduces Robust Graph Diffusion Model (RGDM), a diffusion-based framework for synthesizing node-level graph structures (SGS), including new synthetic nodes and their corresponding edges, within an existing graph to improve robustness of graph neural networks (GNNs) against adversarial attacks. RGDM couples a Robust Graph Autoencoder (RGAE) with a Latent Diffusion Model (LDM). Specifically, the RGAE encodes node attributes and edges into latent features and introduces (i) an edge-selection mechanism to filter adversarial edges and (ii) a low-rank regularization inspired by the Low-Frequency Property (LFP) to denoise features. The LDM then generates latent synthetic structures conditioned on class labels, which are decoded through a new Sparse Hierarchical Edge Decoder (SHED) to produce synthetic nodes and edges efficiently. By merging these synthetic graph structures with the attacked original graph, GNNs trained on the augmented graphs exhibit improved robustness on benchmark datasets (i.e., Cora, Citeseer, Pubmed, Polblogs) under Metattack, Nettack, and Topology Attack settings.

**Strengths:**

1. Synthesizing labeled subgraph structures (nodes + edges) within a graph, rather than generating whole graphs, is a new problem
2.  Empirical results under multiple adversarial attacks and datasets demonstrate its effectiveness on robustness.
3. The experiments are extensive.

**Weaknesses:**

1. Unclear definition and motivation of robustness. It is unclear what “robustness” means in the context of the proposed graph diffusion model. After reading the abstract and introduction, the main motivation appears to be that no diffusion-based approaches have been developed to synthesize graph structures within an existing graph. However, this rationale alone does not explain why robustness is a central issue for such synthesis or what type of robustness is being pursued. I suggest the authors should largely revise these sections to clarify the problem formulation, the notion of robustness targeted, and why it matters beyond novelty.
2. Limited novelty of the low-rank regularization. The proposed low-rank regularization largely borrows from prior low-rank adversarial defense methods [1]. It remains unclear whether the observed robustness improvement truly comes from the diffusion-based synthesis or mainly from this low-rank filtering effect.
3. Insufficient scalability analysis. Although Appendix F.4 briefly reports training and generation runtimes. The results on large-scale graphs are missing. The paper should evaluate performance and efficiency on larger datasets (e.g., OGB-Arxiv or Reddit) to demonstrate that the method can handle realistic large-scale graphs.
4. The paper’s structure makes it difficult to follow. Key motivations are fragmented across multiple sections, and readers must frequently switch between the main text and appendices to follow arguments (e.g., the explanation of Theorem 3.1 is in Appendix B, while its proof is in Appendix A). Likewise, Section 4 repeatedly refers readers to the appendix for essential experimental details that should be included in the main text. A substantial reorganization is recommended to improve readability and narrative flow.
5. In addition, the content is too dense. The authors try to mention too many contents in both main text and appendix but lack many essential details, such as the point 1 I mentioned.
6. Lack of intuition for Theorem 3.1. Theorem 3.1 is presented abruptly without sufficient intuition or discussion in the main paper. The authors should provide conceptual explanations or empirical evidence linking this theoretical result to the observed robustness improvements.
7. Missing threat model specification. Since the paper focuses on improving the robustness of GNNs, a clear description of the attack settings is essential. The threat model is not explicitly stated in the current version and should be clearly defined.
8. Some typos remaining. Line 262: he term -> The term.
9. This paper makes me feel that it aims to integrate many existing techniques, such as diffusion-based generation, edge sparsification, and low-rank regularization together, but too many contents are compressed into one paper.

**Questions:**

Please see the weakness.

---

### Official Review · Reviewer_NLwe · 2025-11-01

**Soundness:** 2
**Presentation:** 2
**Contribution:** 2
**Rating:** 2
**Confidence:** 3

**Summary:**

The authors create RGDM (Robust Graph Diffusion Model), a framework which integrates the Robust Graph Autoencoder (RGAE), Low-Rank Regularization, and Conditional Generation within Latent Space in order to build a solid performance-enhancing framework when handling various forms of GNN-focused adversarial attacks. The authors included details motivations and theoretical for practical components of the framework. Baseline results and an ablation study indicate a solid-level of promise for RGDM on the semi-supervised node-classification task under adversarial perturbations.

**Strengths:**

1) The authors are thorough and clear in the level of detail that they provide for the framework and theoretical justifications and motivations. I appreciate that considerations for Figure 2 and thought that the mix of insights from previous works along with theoretical work was interesting.
2) The effect of RGDM is verified through numerous experiments on Metattack, Nettack, and Topology Attack as well as various citation and co-purchasing networks.
3) Figure 5 is an interesting extension to Figure 1, further discussion or evidence will provide interesting insights into the development of graph-diffusion purification methods.

**Weaknesses:**

1) The writing in the paper is dense, although the technical discussion is interesting from a practical standpoint. It crowds out any potential for further demonstrating the impact or the "why RGDM?" questions. The technical notation is certainly important but it seems that Figure 2, Algorithms 1 and 2 in the appendix do a sufficient point of the mechanism behind RGDM.
2) On the "why RGDM?" -- The authors do make a comparison to other graph-diffusion purification remedies for GNN attacks, noting that the other methodologies are not designed for the node-level task. These seems to serve as the basis for not including any of the methods in their assessment, which seems to form a blind-spot between the contributions of RGDM and what other graph-diffusion purification methods are capable of. More clarification on this is critical to highlight the contribution of this work, since it seems just a question of scope to transfer from graph-based node and structural features to node-based structural features.
3) The results are spliced between the main paper section and the appendix section. The authors note how RGDM works on large-scale graphs and the Topology attack within the main section of the paper but the results for Topology Attack (Table 7) and ogbn-arxiv, Reddit 9 (Table 15) are only mentioned within the appendix. A multitude of results provides a solid basis, but this affects the clarity of the authors claims.

**Questions:**

* Regarding Weakness (1) Why make this distinction? Is it non-trivial to transfer between graph to node features in terms of other diffusion-based purification methods? As far as I am aware, models like the cited DiGress paper [1] are capable of modelling the structural features for nodes and edges (even if they are one-hot). Do you mean the node or edge feature vector itself? If so, then why not apply GraphMAE(2) [2] as a baseline within testing to indicate practical benefits of RGDM?
* Regarding (Weakness 2) These regard my points about "why RGDM?", effectively asking what is important about RGDM's technical contribution. Results which indicate RGDM provides meaningful benefts of self-supervised models could certainly demonstrate this but I am genuinely confused about the practical benefit that RGDM may have over simpler methods. Does its use of TNN have an enhanced upper-bound that is tighter than self-supervised masking? Why is the reconstructed neighbor map being sparse important for the motivation of low-rank regularization? Do we see behavior that is robust to higher than 25% of adversarial noise? Does it provide verifiable protection against sophisticated attacks, like what might be seen in [3]? My confusion mostly stems from how this paper is missing connections to relatively significant papers on graph adversarial perturbations [3] and self-supervised learning [2], which are both deeply-connected to this paper's topic. If the authors can clarify, this will greatly impact my review score.
* Regarding Weakness (3) I understand that the Table 7 and 15 may not have been included due to space limitations. However, why not aggregate results to show RGDM's practical gains across settings? The raw results tables could be retained within the appendix and space could be conserved for more discussion within the main body of the paper.

[1] Vignac, Clement, et al. "Digress: Discrete denoising diffusion for graph generation." ICLR 2023..
[2] Hou, Zhenyu, et al. "Graphmae2: A decoding-enhanced masked self-supervised graph learner." Proceedings of the ACM web conference 2023. 2023.
[3] Dai, Enyan, et al. "Unnoticeable backdoor attacks on graph neural networks." Proceedings of the ACM Web Conference 2023. 2023.

---

### Official Review · Reviewer_nsoB · 2025-11-09

**Soundness:** 3
**Presentation:** 3
**Contribution:** 3
**Rating:** 6
**Confidence:** 3

**Summary:**

The authors propose a new generative approach to improve the robustness of Graph Neural Networks (GNNs) against adversarial attacks on structured data.
The method consists of two main components.
In the first part, called the Robust Graph Autoencoder (RGAE), an encoder–decoder architecture, learns noise-robust node representations. The encoder employs a learnable edge selection mask to filter out corrupted edges and a low-rank regularization to reduce the complexity of the embeddings. The associated decoder reconstructs the graph topology by separately modeling inter- and intra-cluster node connections.
In the second part, the authors train a diffusion model in the latent space of the RGAE, which learns the distribution of “clean” nodes and can generate new synthetic graph structures consistent with the original topology.
Finally, the authors test the method by generating synthetic nodes and edges to be integrated into an attacked graph, producing an augmented graph on which several GNNs are retrained.
Experiments show a significant improvement in robustness compared to existing defense methods.
The paper also introduces two new evaluation metrics, named Frechet Node Distance (FND) and Frechet Edge Distance (FED), to measure the similarity between real and generated nodes and structures.

**Strengths:**

I find the proposed generative defense paradigm for GNNs innovative.
Building the defense mechanism on a generative process makes the approach original compared to traditional defense strategies.
The reasoning behind each architectural component is clear and contributes to a solid overall design, even though I did not verify all formulas in detail.
The experimental results are generally convincing and consistent with the stated goals.

**Weaknesses:**

- The proposed method is quite complex, involving multiple steps and architectural components.
This level of detail makes the paper dense and, at times, difficult to follow.
- Regarding the Sparse Hierarchical Edge Decoder (SHED):


++ The authors do not clearly explain how the number of clusters k used in the decoder is chosen.


++ They state “partitioning the nodes into K balanced clusters using balanced K-means clustering based on node attributes”. However, it is not clear whether the features used are the original node attributes or the latent embeddings learned by the model. If the original (possibly corrupted) features are used, the resulting clusters may inherit noise, degrading the reconstruction quality and contradicting the idea of operating in a “clean latent space.”

- The title “Robust Graph Diffusion Model” is potentially misleading, as it suggests a fully generative model capable of producing entire graphs, while the proposed method is closer to a data augmentation or graph purification technique based on a generative process.
The generation concerns only part of the graph (new nodes and edges).
The approach seems mainly designed to clean and reinforce attacked graphs, rather than to generate new graph instances from scratch.

- The authors list the introduction of Frechet Node Distance (FND) and Frechet Edge Distance (FED) among the main contributions, but the corresponding experiments are placed in the appendix. If these metrics are indeed key contributions, their results should be presented and discussed in the main paper.

- Concerning FND, it is unclear what the authors mean by “node features extracted by a pre-trained GCN.” That is, it is not clear whether the “real” node embeddings refer to nodes from the clean graph or from the attacked graph.

- Since the generative model is trained in the latent space of the RGAE, is there not a direct dependence between the embeddings used for training and for evaluation?
This could introduce a bias in the FND/FED estimation.

- The comparisons with competitors slightly lack clarity.
It is unclear whether the baselines include data augmentation or purification methods, and whether the comparisons are made consistently at the node or graph level.
Also, most of the descriptions of competing methods, e.g., HANG (Zhao et al., 2023), are relegated to the appendix but should appear in the main text to provide proper experimental context.

- I appreciate the large number of experiments conducted, many of which are reported in the appendix.
However, dedicating almost half a page in the main text solely to listing the extra experiments breaks the flow of reading.
It would have been better to integrate this information more smoothly or place it elsewhere.
For instance, the note “The statistical significance analysis of the results in Section 4.2 is performed in Section F.11” could have been moved directly into Section 4.2 for a more coherent presentation.

**Questions:**

- In the encoder section, the update formula includes X’_i W inside the neighbor summation, while one would expect a contribution from j. Is this correct or a typographical error?

- See above

---

### Note · Authors · 2025-12-30

I have read and agree with the venue's withdrawal policy on behalf of myself and my co-authors.